# Uncovering the Impact of Urban Functional Zones on Air Quality in China

Lulu Yuan[1,#], Wenchao Han[2,#,*], Jiachen Meng[3], Yang Wang[1,*], Haojie Yu[1], Wenze Li[2]

[1]College of Earth and Environmental Sciences and College of Atmospheric Sciences, Lanzhou University, Lanzhou, China

[2]State Key Laboratory of Environmental Criteria and Risk Assessment, Chinese Research Academy of Environmental Sciences, Beijing, China

[3]Emergency Management College, Nanjing University of Information Science & Technology, Nanjing, China

[#]These authors contributed equally to this work.

[*]*Correspondence to*: Wenchao Han (han.wenchao@craes.org.cn) and Yang Wang (wang_yang@lzu.edu.cn)

**Abstract.** This study presents a comprehensive spatiotemporal analysis of air quality across various urban functional zones in China from 2017 to 2022, uncovering distinct impacts on air quality due to the unique characteristics of each zone. A general decrease in various pollutant concentrations is observed, a result of stringent pollution control policies. Specifically, the concentration of $PM_{2.5}$ decreased from 46.1 μg/m³ to 30.6 μg/m³. Residential, commercial, and industrial zones show significant declines, whereas the transportation zone experiences the least decrease. However, ozone levels rebound

significantly in densely populated residential and commercial zones, and exhibit distinct weekend effects. The research highlights U-shaped seasonal patterns for five key pollutants and inverse seasonal patterns for ozone, which gradually decrease. Furthermore, the daily and seasonal variations of pollutant concentrations in industrial zone are the largest, while those in the public management and service zone are the smallest. For example, the seasonal fluctuation of $PM_{2.5}$ and $PM_{10}$ in industrial zone was 50.5 μg/m³ and 66.1 μg/m³, respectively. Urban scale has the most significant impact on public management and

service zone. Notably, spatial heterogeneity is evident, with regional pollutant distributions linked to local emissions, control measures, urban morphology, and climate variability. This study emphasizes the critical link between urbanization and air quality, advocating for continuous monitoring and the development of zone-specific air quality strategies to ensure sustainable urban environments.

## 1 Introduction

The rapid development of the world economy and the ongoing trend of urbanization have made air pollution an inescapable environmental issue, with effectively managing and reducing it becoming a hotly debated topic across all social classes (Li et al., 2023; Wen et al., 2024; Zhang et al., 2022a). Over long periods, air pollutants interact with radiation, clouds, and water vapor, thereby altering global and regional climates (Fan et al., 2013; Li et al., 2017). Additionally, air pollution triggers a variety of short-term extreme weather events, such as extreme precipitation, floods, droughts, wildfires, and heat waves (He

et al., 2024; Rosenfeld et al., 2008). Numerous studies have demonstrated the close association of air pollution with neurological diseases, cardiovascular diseases, and lung cancer, significantly impacting human health (Berg et al., 2023; Qin et al., 2023; Ward-Caviness and Cascio, 2023). The frequent changes in urban land use and the increased intensity of land use due to urbanization will lead to significant changes in the emissions and dispersion conditions of urban air pollutants, altering the spatial distribution of air pollution within cities (Qi et al., 2022; Qian et al., 2022). Consequently, it is crucial to study how

urban underlying surfaces influence the spatial distribution of air pollutants to enhance our understanding of the interaction between urbanization and air pollution, as well as to improve the accuracy of air pollution control measures.

Situated in Eastern Asia with an expansive landmass spanning 9.6 million square kilometers characterized by diverse topography and notable regional climatic variations (Chen and Sun, 2015; Wang, 2010), China stands as both the world's largest developing nation and leading emitter boasting a substantial agricultural economy and population base (Chen and Gong,

2021). Since 2013, rigorous national pollution standards along with strategic control initiatives like "the Air Pollution Prevention and Control Action Plan" (SC, 2013) and "the Three-Year Action Plan for Winning the Blue Sky Defense Battle" (SC, 2018) have significantly enhanced air quality; however, levels of $PM_{10}$, $PM_{2.5}$, and $O_3$ persist well above World Health Organization benchmarks (Liu et al., 2023; Zeng et al., 2019).

Based on data from remote sensing and station observations, the majority of previous research has examined the regional and

temporal distribution of air pollution in China across various spatial and temporal scales. In terms of temporal scale, numerous studies have analyzed the changing trends of various pollutants in recent years based on inter-annual variations. The findings indicate that the concentration of $O_3$ has been the only one to increase, while the concentrations of other pollutants ($PM_{2.5}$, $PM_{10}$, $SO_2$, $NO_2$, and CO) have decreased year by year (Deng et al., 2022; Fan et al., 2020; Yang et al., 2024). From the perspective of seasonal variation, the concentrations of $PM_{2.5}$, $PM_{10}$, $SO_2$, $NO_2$, and CO exhibit a U-shaped distribution,

characterized by higher concentrations in winter and lower concentrations in summer; conversely, $O_3$ demonstrates a distribution that is inversely U-shaped (Dong et al., 2022; Fang et al., 2022; Wang et al., 2022a). Regarding diurnal variation, the diurnal concentration of $PM_{2.5}$ displays a bimodal trend, whereas $SO_2$ and $O_3$ exhibit unimodal patterns (Liu et al., 2024; Qin et al., 2021; Zhao et al., 2016). In terms of spatial scale, the concentration of pollutants other than $O_3$ reaches the highest value in North China across different regions (Fan et al., 2020; Kuerban et al., 2020). When considering different provinces,

the high concentration areas of particulate matter ($PM_{2.5}$ and $PM_{10}$) are mainly found in Hebei, Henan, Shandong, and Xinjiang provinces; the $SO_2$ concentration is highest in Shanxi Province, the lowest in Hainan Province, and the high concentration area of CO is also in Shanxi Province (Wang et al., 2022a; Zhao et al., 2021a). From the perspective of different urban agglomerations, the $PM_{2.5}$ concentration in the Beijing-Tianjin-Hebei urban agglomeration has decreased the most in recent years, and the high concentration area of air pollutants has gradually shifted southward to the Central Plains urban

agglomerations, and it is observed that pollutant concentration significantly correlates with the level of economic development and population density (Qi et al., 2023; Tao et al., 2022).

There exists a significant correlation between urbanization and air pollution. Airborne pollutants have the ability to disperse and absorb solar radiation, leading to a reduction in ground-level solar radiation intensity (Fan et al., 2018a; Zhang et al., 2022b). On the other hand, air pollutants can affect the relative humidity of clouds, leading to changes in cloud thickness and coverage, and the radiation from clouds also changes accordingly, which in turn affects the rainfall and thunderstorms in urban

areas (Guo et al., 2016; Wilcox et al., 2016). Furthermore, air pollutants can also alter the intensity of urban heat islands by changing the atmospheric temperature above the city, altering the stability of the boundary layer and vertical movements, as well as reducing diurnal temperature fluctuations (Cao et al., 2016; Han et al., 2020; Youn et al., 2023).

Changes in anthropogenic emissions and the spatial structure of urban underlying surfaces caused by urbanization will have

an important impact on the formation, transmission, and distribution of air pollutants in urban areas (Ban et al., 2023; Shen et al., 2017; Yang et al., 2020). Pollutant discharge has a direct impact on the concentration levels and spatial distribution of pollutants in the atmosphere (Li et al., 2022a; Zawacki et al., 2018). Changes in the spatial structure of urban underlying surfaces not only change the spatial pattern of pollutant emissions but also affect the diffusion conditions of pollutants through changes in urban heat island effect (UHI) and urban canopy meteorological forcing (UCMF), affecting the spatial distribution

of air pollutants (Fan et al., 2018b; Huszar et al., 2022; Zhong et al., 2018). Based on urban land use data and built-up area data, many studies have characterized the spatial structure of the underlying urban surfaces through parameters such as urban compactness index and landscape pattern index, and have found that urban form and urban scale affect the distribution and diffusion of air pollutants, especially in $PM_{2.5}$, $NO_2$, and $O_3$ (Chen and Wei, 2024; Huang et al., 2021a; Liu et al., 2022; Tao et al., 2020; Zhang et al., 2022c). Some studies have found that the impact of urban landscape form on air quality varies according

to city size and location, and generally small-scale, decentralized, and multi-center cities have better air quality (Li and Zhou, 2019; Mao et al., 2022; Zhao et al., 2022; Zhu et al., 2023). The influence of urban landscape patterns on pollutant concentrations has also shown significant spatial heterogeneity. Patch density (PD) in Henan and Shandong provinces showed

a significant positive correlation with PM$_{2.5}$ concentrations, while the urban PD in Northeast China showed a significant negative correlation with PM$_{2.5}$ concentrations. The correlation between landscape patterns and PM$_{2.5}$ concentrations in inland and coastal cities is also significantly different. The landscape in the built-up area of Hohhot has a significant positive effect on PM$_{2.5}$ concentrations, and the landscape in green space has a significant inhibitory effect on PM$_{2.5}$ concentrations, but this effect is not as obvious in Tianjin. The correlation between landscape patterns and PM$_{2.5}$ concentrations also varies within different areas of the same city. The density of industrial buildings in suburban Shenzhen is significantly positively correlated with PM$_{2.5}$ concentrations, while in downtown Shenzhen it is negatively correlated (Duan et al., 2021; Shen et al., 2023; Wang et al., 2019). In addition, the influence of urban landscape patterns on PM$_{2.5}$ concentrations also has seasonal and scale effects, which is significant in summer at the regional scale and even more so in winter at the urban grid scale (Li et al., 2021; Meng et al., 2023).

Although numerous studies have been conducted on the relationship between urbanization and air pollution, no relevant research has been found regarding the impact of urban functional zones on air quality. The significant differences in emissions between various urban functional zones, such as residential, industrial, and transportation areas, coupled with the distinct types of spatial structures of underlying surfaces, lead to divergent impacts on the evolution of atmospheric pollutants across these zones. Nevertheless, the specific influence of various urban functional zones on air pollution dynamics remains unclear. A comprehensive analysis of the spatio-temporal evolution of urban air quality in China from the perspective of urban functional zones can offer novel insights and methodologies for preventing and controlling urban air pollution, as well as guiding future urban planning and environmental management practices.

Therefore, drawing upon extensive ground station observation data spanning several years, this study aims to conduct a systematic and comprehensive analysis of the spatio-temporal evolution characteristics of air pollutants across various functional zones within Chinese urban areas from 2017 to 2022. This analysis will encompass multiple temporal scales (yearly, seasonal, weekly, daily) as well as diverse spatial dimensions (geographical regions and urban agglomerations), while also examining their potential influencing factors. The specific research design is as follows: the second chapter is the data and methods, the third chapter is the results and discussion, and the fourth chapter is the conclusion.

## 2 Data and methods

### 2.1 Study area

China is located in the eastern part of Asia, on the western coast of the Pacific Ocean, with a large span of latitudes. It has a vast territory and a terrain that is undulating and varied, which can be roughly divided into three levels of steps, with the altitude gradually decreasing from west to east. The complex and diverse topography has led to a complex and diverse range of climate types (Cheng et al., 2018; Liu and Liu, 2023). The climates in China include tropical monsoon climate, subtropical monsoon climate, temperate monsoon climate, temperate continental climate, and plateau mountain climate. As the globe's foremost consumer of resources and the second-largest economy, China has witnessed remarkable economic expansion over recent decades, propelled by its abundant human capital and ongoing industrialization processes (Dmitrienko et al., 2023; Guo et al., 2013; Zhang et al., 2023). This study divides China into six major regions based on geographical characteristics and administrative divisions: North China, Northeast China, East China, South-Central China, Southwest China, and Northwest China.

In this study, 336 prefecture-level cities in China are selected as research objects, which have great differences in natural conditions, economic development, and industrial structure. As the scale of Chinese cities continues to expand and the economic strength of cities continues to increase, the Chinese government promulgated the National New Urbanization Plan (2014–2020) in 2014, which proposed a "5-8-6" pattern of urban agglomerations, namely 5 national urban agglomerations, 8 regional urban agglomerations, and 6 local urban agglomerations. The 14th Five-Year Plan further emphasizes the promotion

of 19 urban agglomerations (He et al., 2022; Ouyang et al., 2021). This study will focus on six major urban agglomerations

(Figure 1): Beijing-Tianjin-Hebei urban agglomeration (BTH), Yangtze River Delta urban agglomeration (YRD), Triangle of Central China (TC), Greater Bay Area urban agglomeration (GBA), Chengdu-Chongqing urban agglomeration (CC), and Northern Slope of Tianshan Mountains urban agglomeration (NSTM).

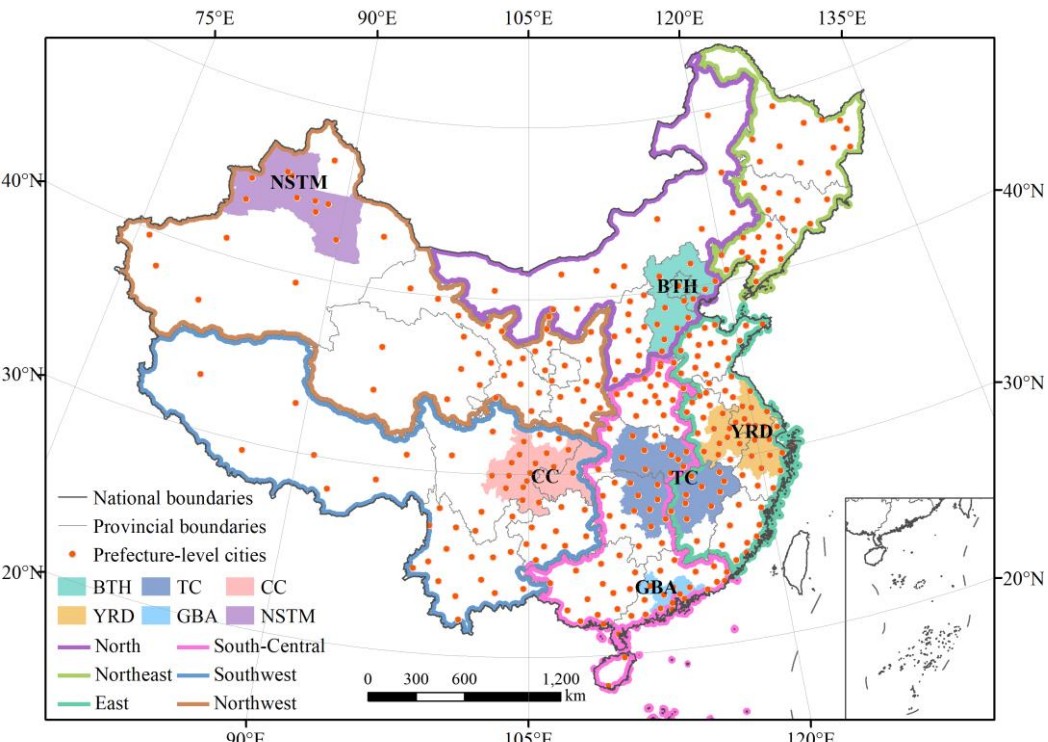

**Figure 1.** The location of study area and the distribution map of six geographical regions and six major urban agglomerations. BTH: Beijing-Tianjin-Hebei urban agglomeration; YRD: Yangtze River Delta urban agglomeration; TC: Triangle of Central China; GBA: Greater Bay Area urban agglomeration; CC: Chengdu-Chongqing urban agglomeration; NSTM: Northern Slope of Tianshan Mountains urban agglomeration.

**2.2 Data sources**

The hourly observation data of pollutants from national air quality control stations during 2017-2022 used in this study were derived from urban air quality monitoring data released by the Ministry of Ecology and Environment of China (https://quotsoft.net/air/). The data covers 1,482 sites across 336 prefecture-level cities in China and includes hour-by-hour mass concentrations of six pollutants (PM$_{2.5}$, PM$_{10}$, SO$_2$, NO$_2$, O$_3$, and CO).

The urban functional zone data comes from the 2018 China Basic Urban Land Use Type Map (EULUC-China) dataset (https://data-starcloud.pcl.ac.cn/iearthdata/). This map is based on the comprehensive use of 30-meter city contour data, OpenStreeMap road network data, multi-source remote sensing data, night light data, and social big data (Autonavi POI number and Tencent positioning population change characteristics) to determine the classification characteristics of urban plot scale, and the random forest algorithm is used to draw. China's cities are divided into five functional zones: residential zone,

commercial zone, industrial zone, transportation zone, and public management and service zone (Gong et al., 2020).

The national Digital Elevation Model (DEM) data comes from the global DEM dataset released by the General Bathymetric Chart of the Oceans (GEBCO) in 2022 (https://www.gebco.net/data_and_products/). This dataset includes global DEM data from grid scale to basin scale, covering sea level change and ocean topography, etc. Grid DEM is combined with high-resolution satellite remote sensing images, and the global land and ocean areas are divided. DEM data is raster data with a

spatial resolution of 500 meters.

The population data is obtained from the LandScan Global dataset developed by Oak Ridge National Laboratory (ORNL) in

2020 (https://landscan.ornl.gov/). This dataset integrates geospatial science, remote sensing technology, and machine learning algorithms to estimate global population distribution at a spatial resolution of 1 km. The LandScan Global algorithm uses spatial data, high-resolution imagery exploitation, and a multi-variable dasymetric modeling approach to disaggregate census counts within an administrative boundary.

## 2.3 Data preprocessing

The hourly concentration data of pollutants in the site were screened and cleaned, and the abnormal values and missing values in the data were eliminated. The minimum validity requirements of pollutant concentration data stipulated in Ambient Air Quality Standards (GB3095-2012) (https://www.mee.gov.cn/) shall be strictly followed during data processing, and data quality control shall be adopted to ensure the validity and reliability of atmospheric pollutant concentration data. Specific methods are as follows (Li et al., 2019a; Silver et al., 2018; Yu et al., 2024):

1) Any concentration per hour value that is missing or less than or equal to zero is considered invalid data.

2) Each monitoring site should have an effective concentration value for at least 20 hours in a day. If a site has less than 20 active hours on a given day, the data for that day is considered invalid.

3) Each monitoring site should have at least 27 effective daily average concentration values in a month.

4) Each monitoring site should have at least 324 effective daily average concentration values in one year.

5) If the $PM_{2.5}$ concentration of a station is greater than $PM_{10}$ for an hour, the data for that hour will be considered invalid.

## 2.4 Data analysis process

### 2.4.1 Overlay Analysis

Overlay analysis is a spatial analysis technique that involves combining multiple geographic layers to reveal spatial relationships and attribute associations between different elements. In general, overlay analysis is mainly used to integrate various types of spatial data, identifying intersecting, union, or difference areas to provide a foundation for subsequent analyses. Common operations include Intersect, Union, Erase, and Spatial join. In this study, ArcGIS was employed to overlay the latitude and longitude data of the sites with the urban functional area data, allowing for the identification of the functional area category for each site through spatial connections. This process facilitated the addition of new functional area attribute information to the site data. Using a similar methodology, the longitude and latitude data were overlaid with DEM grid data to obtain elevation information for each station. This integration of spatial and attribute data not only provided a more nuanced understanding of the spatial distribution of the sites but also laid the groundwork for further investigations into the relationships between site characteristics and urban functional zones.

### 2.4.2 Data statistics

In this study, MATLAB was employed as the primary computational tool to develop a customized code for batch processing the pre-processed pollutant concentration data obtained from various monitoring sites. This approach facilitated the efficient calculation of daily, monthly, and annual mean values for six key pollutants ($PM_{2.5}$, $PM_{10}$, $SO_2$, $NO_2$, $O_3$, and CO) at each site, thereby providing a comprehensive temporal overview of pollutant concentrations (Fan et al., 2020). Following the temporal analysis, the study proceeded to spatially categorize the data based on functional area classifications. The specific operation method is to establish the site index for each functional area. The corresponding pollutant concentration data for each functional area was extracted based on this site index, leading to the classification of pollutant concentration data across different functional zones. Finally, the average concentrations and variation trend of the six pollutants within each functional area were computed across various spatial scales, including six geographical regions, six urban agglomerations, and different altitudes. The variation trend of pollutant concentrations was characterized by the relative rate of decline, with the specific formula presented as follows.

$$Trend\left(\%\right) = \frac{\left(x_{2022} - x_{2017}\right)/5}{\sum_{i=2017}^{2022} x_i / 6} \times 100\%$$

where $i$ represents the year, $x$ represents the average concentrations of six pollutants.

## 3 Results and discussion

### 3.1 Number of sites in different functional zones

Following the superposition analysis of the functional zone and site latitude and longitude data, it is discovered that 118 sites are unrelated to any functional zone and that 1364 sites in total overlap with the functional zone. 1364 sites that coincide with functional zones are thus chosen for analysis in this research. In general, the majority of the sites are concentrated in public management and service and residential zone; of them, 555 are in public management and service zone , 320 are in residential zone, and just 69 are on transportation zone (Table 1).

From the perspective of the six geographical regions, it is evident that the East and South-Central regions contain a relatively high number of stations. To be specific, these two regions are home to 176 and 132 residential zone stations respectively. In contrast, the Northwest region has a significantly smaller number of stations, with only 49 residential zone stations. In terms of the six major urban agglomerations, the Yangtze River Delta urban agglomeration and the Triangle of Central China boast a relatively large number of stations. Conversely, the Northern Slope of Tianshan Mountains urban agglomeration has the fewest stations, with each urban functional zone within it having fewer than 10 stations. Furthermore, among all the urban agglomerations, the Chengdu-Chongqing urban agglomeration has the highest number of transportation zone stations.

Table 1. Statistics on the number of monitoring stations per urban functional zone per region, and per urban agglomeration.

|  |  | Residential | Commercial | Industrial | Transportation | Public management and service |
|---|---|---|---|---|---|---|
| Six geographical regions | North | 62 | 23 | 35 | 5 | 46 |
|  | Northeast | 63 | 15 | 36 | 12 | 32 |
|  | East | 176 | 38 | 71 | 13 | 89 |
|  | South-Central | 132 | 34 | 75 | 22 | 73 |
|  | Southwest | 73 | 17 | 25 | 16 | 42 |
|  | Northwest | 49 | 15 | 36 | 1 | 38 |
|  | Total | 555 | 142 | 278 | 69 | 320 |
| Six urban agglomerations | BTH | 31 | 6 | 12 | 4 | 21 |
|  | YRD | 76 | 8 | 27 | 3 | 42 |
|  | TC | 66 | 11 | 21 | 2 | 33 |
|  | GBA | 23 | 3 | 7 | 2 | 19 |
|  | CC | 37 | 2 | 10 | 7 | 18 |
|  | NSTM | 9 | 0 | 9 | 1 | 7 |
|  | Total | 242 | 30 | 86 | 19 | 140 |

### 3.2 Temporal variation

### 3.2.1 Overall interannual variation

The annual variation of $PM_{2.5}$, $PM_{10}$, $SO_2$, $NO_2$, and CO concentrations in various functional zones of Chinese cities is illustrated in Figure 2 and Figure S1. From a nationwide perspective, there has been a consistent year-on-year decline in the concentrations of these pollutants from 2017 to 2022. Specifically, the concentration of $PM_{2.5}$ decreased from 46.1 μg/m³ to 30.6 μg/m³, which corresponds to a percentage decrease of 33.7% ($P < 0.01$). Compared with other pollutants, the concentration of $SO_2$ has the largest decrease, with a decrease rate of up to 53.5% (Figure 2c). This is due to China's emission reduction measures for air pollution control and loose coal control in recent years, which led to a continuous decline in $SO_2$ emissions (He et al., 2023a; Huang et al., 2021b). However, $O_3$ concentration showed a downward trend from 2018 to 2021, and then rebounded in 2022. Compared with 2017, $O_3$ concentration decreased by 5.6% in 2021 and increased by 3.7% in 2022 (Figure 2e). In recent years, $NO_x$ emissions in China have decreased significantly, while VOCs have decreased slightly. Due to the highly nonlinear response relationship between $O_3$ and its precursors $NO_x$ and VOCs, coupled with the influence of meteorological conditions, ozone concentration has increased (Lu et al., 2021; Wang et al., 2013; You et al., 2017).

Significant variations are observed in the reduction of pollutant concentrations across different functional zones. Except for $O_3$, the concentration of pollutants has improved most significantly in residential, commercial, and industrial zones. Compared with 2017, $PM_{2.5}$ concentration had decreased by 34.3%, 35.5%, and 33.8% in these areas by 2022, and $SO_2$ concentration had decreased by 55.6%, 56.4%, and 53.4%, respectively ($P < 0.01$). The main reason is that following the implementation of policies such as the "Clean Winter Heating Plan for Northern China (2017–2021)" (NDRC, 2017) and the "Three-Year Action Plan for Winning the Blue Sky Defense Battle", the government intensified control over industrial pollution, promoted clean production in enterprises, implemented clean heating measures in residential areas, and encouraged the use of clean fuels like natural gas and electricity (Song et al., 2023; Wang et al., 2022b). The power and industrial sectors have registered significant achievements in reducing emissions and have taken a leading role in driving phenomenal improvements in air quality (Geng et al., 2024). However, the rate of decrease in pollutant concentration in transportation zone was significantly lower than in other zones, with $NO_2$ showing the least reduction at 28.3% and an average annual decrease of 1.72 μg/m³ ($P < 0.01$), indicating the slowest improvement. Road traffic is the primary source of $NO_2$ pollution (Xin et al., 2021). Although China has continued to implement initiatives to transform to cleaner automobiles to reduce traffic emissions of $NO_2$, further efforts are required. In contrast to other pollutants, $O_3$ concentration has seen a significant rebound in residential, commercial, and transportation areas with high population activity, with an increase rate of 5.53%, 3.97%, and 4.01% in 2022, respectively, while in industrial areas, the rebound was minimal at only 2.64%. An increasing number of studies have shown that long-term exposure to $O_3$ pollution can adversely affect human health, potentially causing diseases of the nervous and respiratory systems, and even leading to premature death (He et al., 2023b; Li et al., 2022b; Yim et al., 2019). Therefore, reducing ozone concentration, particularly in densely populated areas, is imperative.

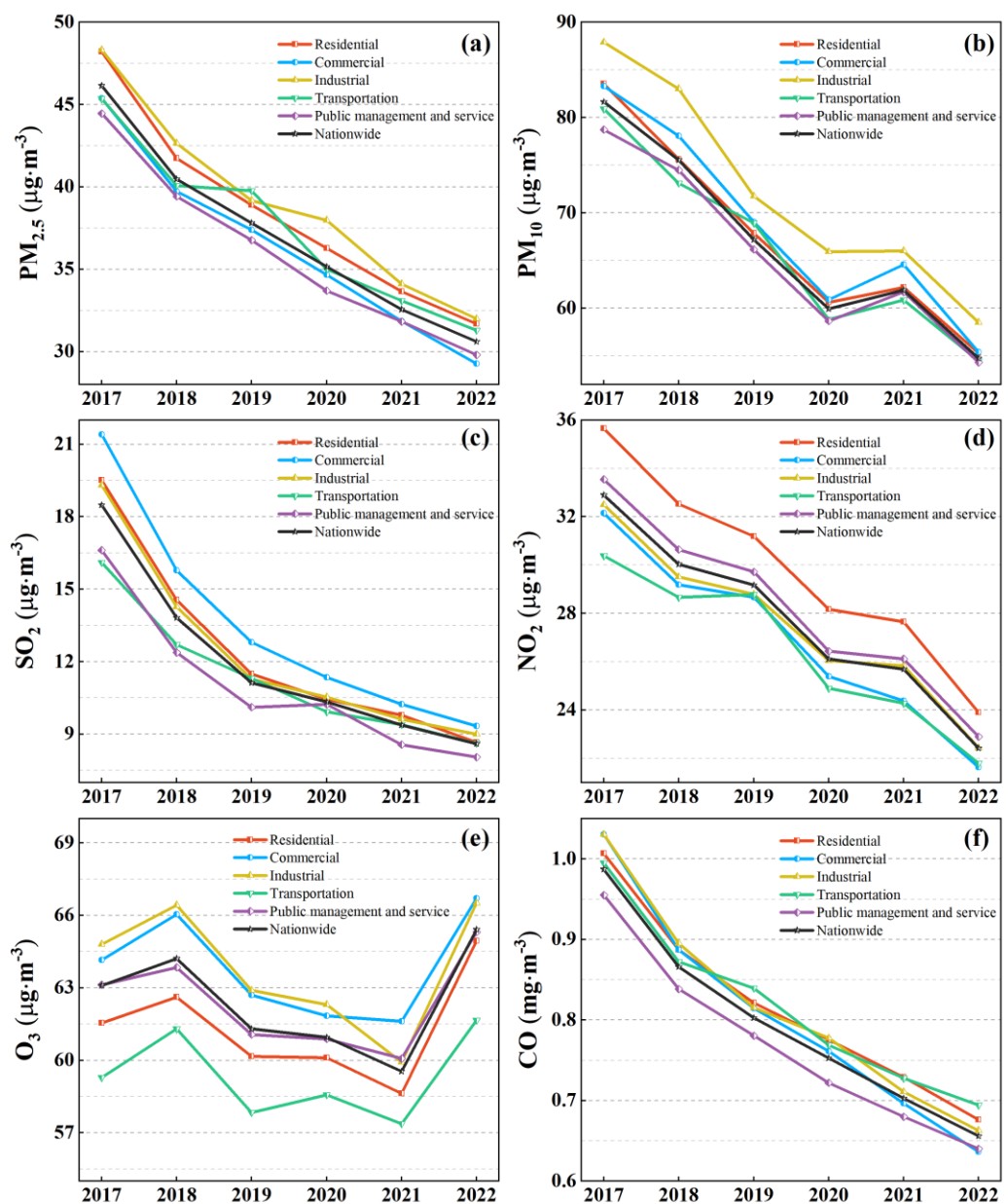

**Figure 2.** Annual variation of six pollutant concentrations in various functional zones of Chinese cities. (a) PM$_{2.5}$, (b) PM$_{10}$, (c) SO$_2$, (d) NO$_2$, (e) O$_3$, (f) CO.

### 3.2.2 Seasonal and daily variation

Based on the national seasonal changes (Figure 3 and Figure S2), the concentrations of PM$_{2.5}$, PM$_{10}$, SO$_2$, NO$_2$, and CO exhibited a U-shaped distribution from January to December, with higher concentrations observed in winter compared to summer. The high concentration of PM$_{2.5}$ in winter in China can be attributed to coal-burning heating in northern China, and meteorological conditions that are not conducive to the diffusion of pollutants (winter temperature inversion, more stable atmospheric conditions, less wet deposition) also contribute to the accumulation of air pollutants (Fan et al., 2021; Wang et al., 2017). In contrast to other pollutants, O$_3$ displayed an inverted U-shaped seasonal variation with higher concentrations in summer than in winter. This finding aligns with previous studies by Wang et al. (2022c) and Fan et al. (2020). The increased atmospheric temperature, intensified solar radiation, and extended sunshine duration during summer facilitate enhanced photochemical reactions leading to heightened O$_3$ generation (Barzeghar et al., 2020).

Overall, the seasonal fluctuations of the six pollutants showed a downward trend from 2017 to 2022. The seasonal fluctuation of SO$_2$ and CO showed a decreasing trend year by year, and the difference of SO$_2$ and CO decreased from 22.8 μg/m³ to 3.9 μg/m³, and from 0.72 mg/m³ to 0.43 mg/m³. This downward trend can be attributed to the adjustment of China's energy structure

and the implementation and improvement of desulfurization technology. In recent years, with the gradual reduction of dependence on high-emission energy (such as coal), the shift to clean energy, and the continuous improvement of energy efficiency, the emission reduction of $SO_2$ and CO has achieved remarkable results, and the seasonal differences have also decreased (Qian et al., 2020). Specifically, $SO_2$ has the most significant decline in commercial zone, with a decrease of 85.2%.

Commercial zone mainly includes shops, restaurants, etc., whose main emission sources may come from the coal or gas of small heating facilities and cooking equipment. After the implementation of environmental protection policies and technological improvements, the $SO_2$ emissions of these emission sources have been significantly reduced. CO emissions from industrial zone experienced the most significant decrease, dropping by 42.4%. These emissions primarily arise from combustion, chemical reactions, and various industrial production processes. Following improvements in cleaner production

methods and emission control technologies, CO emissions have also been further reduced. $NO_2$ and $O_3$ show the least variation in seasonal fluctuation, particularly in areas designated as public management and service zone. This stability can be attributed to the scarcity of pollutant emission sources in public management and service zone, as well as the stringent environmental monitoring and management practices that are implemented.

While there exist certain variations in seasonal fluctuations of pollutants among different urban functional zones, these

275 differences are not significant (Figure S3). For instance, the seasonal fluctuation of particulate matter ($PM_{2.5}$ and $PM_{10}$) in industrial and transportation zones is higher than that in other functional zones. The seasonal fluctuation of $PM_{2.5}$ ($PM_{10}$) in industrial and transportation zones is 50.5 μg/m³ (66.1 μg/m³) and 51.0 μg/m³ (65.7 μg/m³), respectively. The primary cause of these seasonal changes is the variation in particulate matter concentration within the environment, particularly when there is low removal efficiency of industrial dust with high emissions, leading to an increase in this seasonal trend accordingly (Li

et al., 2022c; Luo et al., 2022). Thus, the emission levels of particulate matter from industrial and transportation zones surpass those from other functional zones, resulting in conspicuous seasonal fluctuations. The greater seasonal fluctuation of $NO_2$ and $O_3$ observed in residential zone and industrial zone compared to other functional zones can be attributed to activities such as heating and coal burning during winter, which elevate the production of nitrogen oxides along with ozone precursors within the atmosphere (Huang et al., 2013). Additionally, CO in industrial zone exhibited a larger seasonal fluctuation at 0.59mg/m³

compared to other functional zones. In summary, it can be concluded that the largest variations among various pollutants occur in industrial zone, while public management and service zone experiences minimal fluctuations.

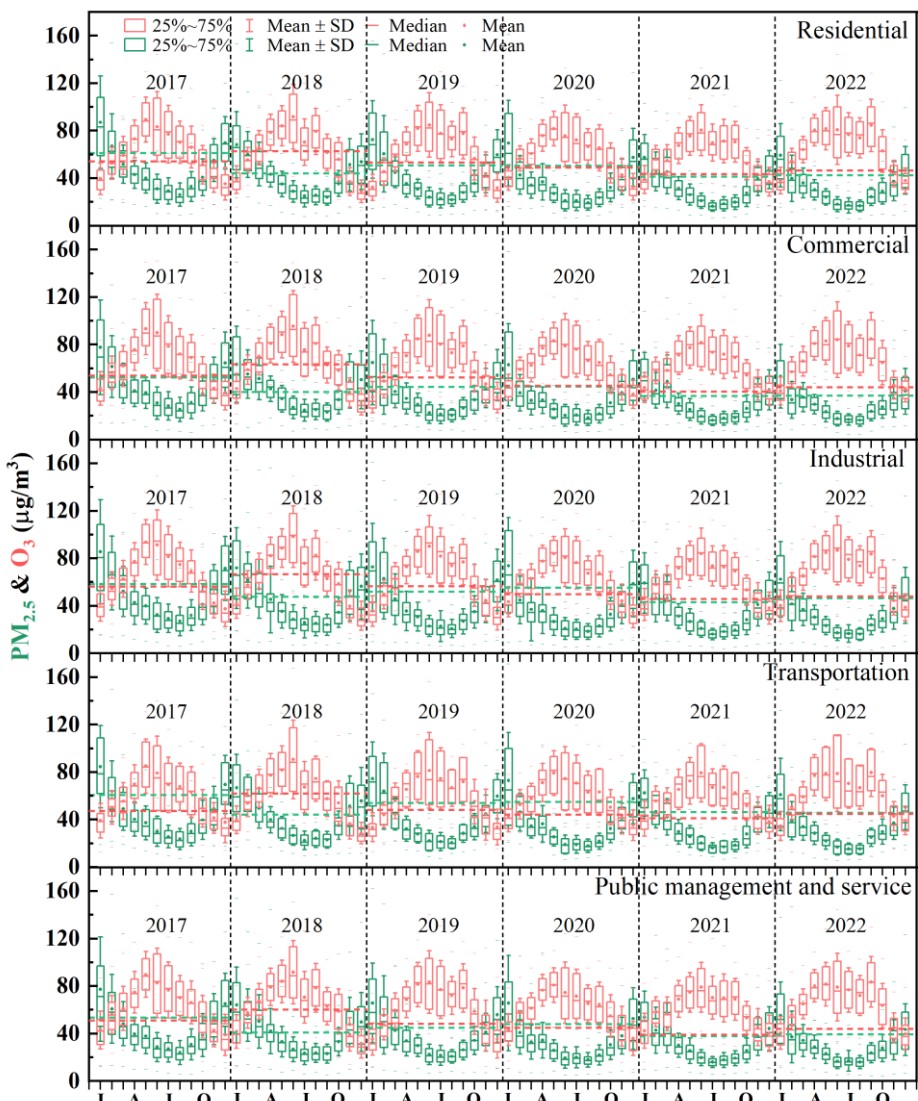

**Figure 3.** Seasonal variation trend of PM$_{2.5}$ and O$_3$ concentrations in various functional zones of Chinese cities. The dashed lines indicate the difference between the highest and lowest monthly mean concentration in the corresponding year.

As depicted in Figure 4 and Figure S4, the daily variation amplitude of different functional zones for NO$_2$ and O$_3$ exhibits minimal differences, significantly lower compared to the other four pollutants. Given that ozone is a regional pollutant, the variations among different functional zones are relatively small (Wang et al., 2022c). The daily variation of particulate matter (PM$_{2.5}$ and PM$_{10}$) in transportation, industrial, and commercial zones is greater than that in residential and public management and service zones. This disparity can be attributed to factors such as working hours and peak traffic periods. Transportation, industrial, and commercial zones, which are heavily trafficked and populated during business hours and peak times, contribute to higher emissions of particulate matter and, consequently, exhibit greater daily variation (Song et al., 2019). Specifically, the daily variation of PM$_{2.5}$ concentrations in public management and service zone stands at 7.8 μg/m³, significantly lower than that observed in other functional zones ($P < 0.05$). In commercial zone, the daily variation of SO$_2$ and CO notably exceeds that of other functional zones, with levels reaching 6.74 μg/m³ and 0.29 mg/m³, respectively ($P < 0.05$). The emissions from the catering industry's cooking processes, peaking during lunch and dinner hours, are a significant source of these pollutants (ElSharkawy and Ibrahim, 2022). Conversely, public management and service zone, which includes government agencies, schools, and hospitals, shows the smallest daily variation in SO$_2$ and CO levels at 3.99 μg/m³ and 0.22 mg/m³, respectively, due to minimal pollutant emissions. Overall, the daily variation of pollutants is most pronounced in commercial, industrial, and transportation zones, while public management and service zone exhibits the least variation.

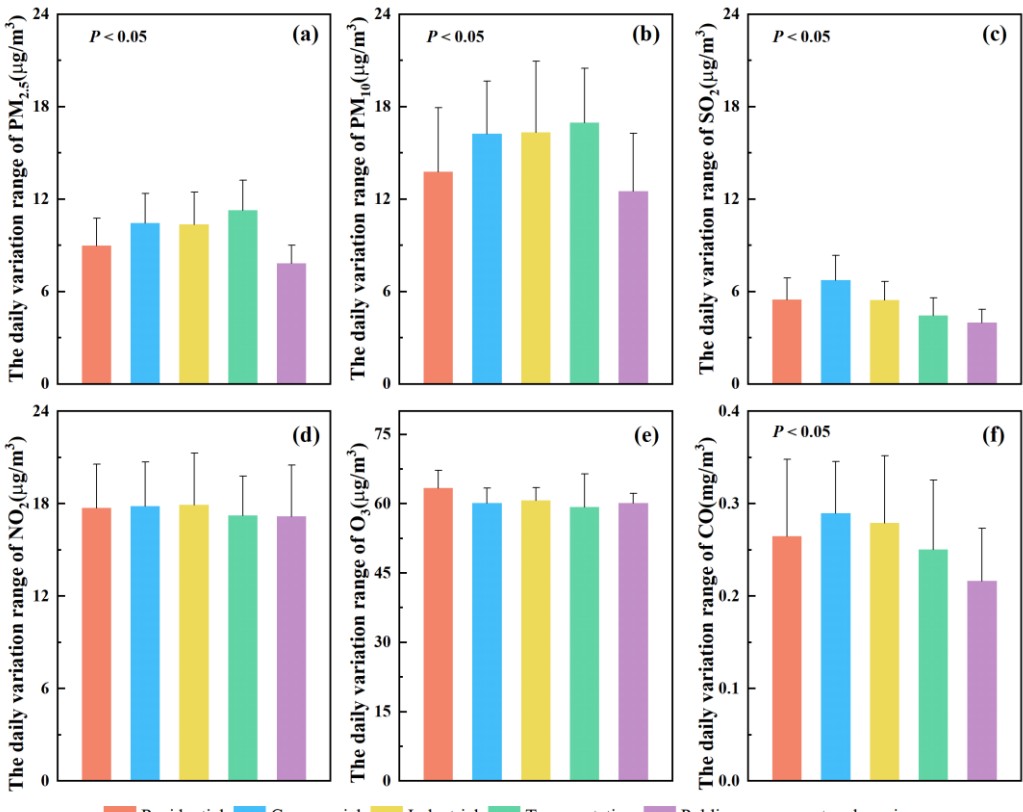

**Figure 4.** Daily variation range of $PM_{2.5}$ (a), $PM_{10}$ (b), $SO_2$ (c), $NO_2$ (d), $O_3$ (e), and CO (f) concentrations in various functional zones of Chinese cities.

### 3.2.3 The weekend effect

The weekend concentration less than the weekday concentration is defined as the "positive weekend effect", and the weekend concentration greater than the weekday concentration is defined as the "negative weekend effect". The concentrations of $PM_{2.5}$, $PM_{10}$, $SO_2$, and $O_3$ on weekends are higher than those on weekdays, showing a " negative weekend effect" (Figure 5 and Figure S5). On the contrary, the concentration of $NO_2$ on weekends is smaller than that on weekdays, showing a " positive weekend effect". The average concentration of CO on weekdays and weekends is not much different, and the "weekend effect" is not obvious.

The phenomenon known as the "weekend effect" varies across distinct functional zones for each pollutant. For particulate matter (PM), a pronounced "negative weekend effect" is observed in residential, industrial, and transportation zones. The increased consumption of coal and biomass for residential living and the high proportion of people traveling on weekends lead to significantly higher particulate matter emissions from human activities on residential zone and transportation zone on weekends than on working days (Hua et al., 2021). In recent years, the influence of the pandemic and the implementation of the industrial "off-peak production" policy have caused certain factories to operate normally on weekends, potentially contributing to the heightened weekend concentrations of PM in industrial zone. Regarding $SO_2$, the "negative weekend effect" is most pronounced in commercial zone, with weekend concentrations exceeding those of weekdays by 1.74%. This could be due to there being more $SO_2$ emissions from catering in commercial zone on weekends. In addition, the "positive weekend effect" for $NO_2$ is most pronounced in public management and service zone, where weekend concentrations are 1.03% lower than those on weekdays. This reduction may be attributed to the fact that public management and service zone predominantly encompasses government institutions and educational facilities, where vehicular traffic and, consequently, $NO_2$ emissions from vehicles, are significantly reduced during weekends (Zheng et al., 2023).

The "negative weekend effect" observed in the $O_3$ concentrations of residential and public management and service zones parcels exceeds that of other functional zones, exhibiting an inverse trend in comparison to $NO_2$ exactly. This phenomenon

corroborates the findings reported by (He, 2023), who identified a pronounced negative correlation between the weekly fluctuation patterns of $NO_2$ and $O_3$. The formation process of $O_3$ in most areas of China is limited by volatile organic compounds (VOCs), and the reduction of $NO_x$ emission can lead to the escalation in $O_3$ levels.

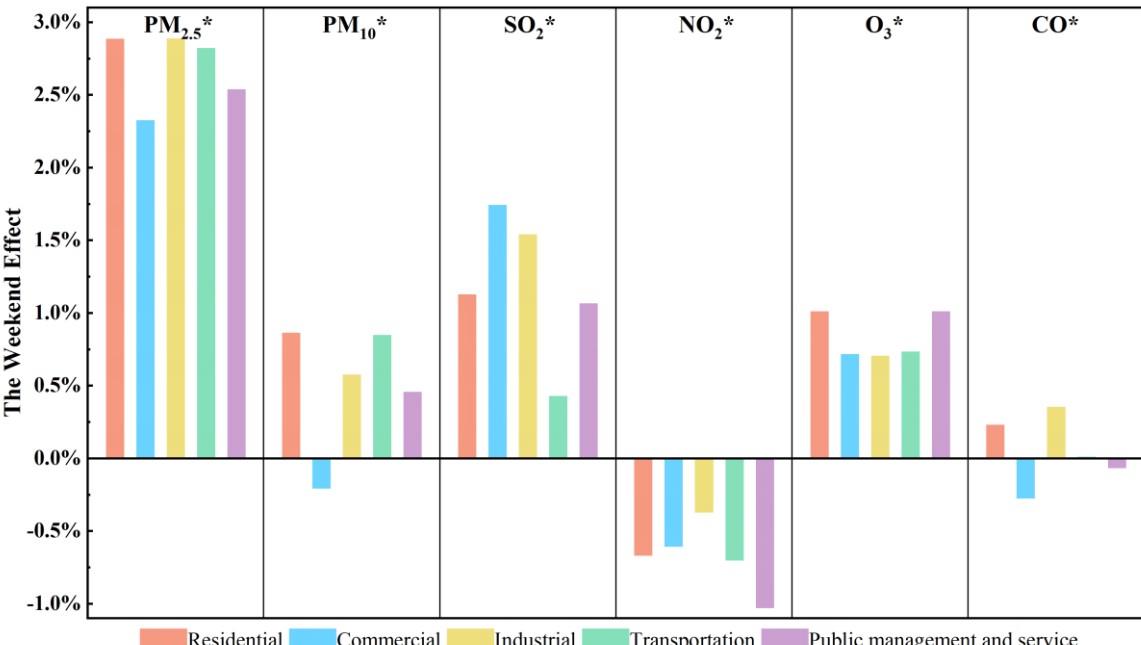

**Figure 5.** Weekend effect((weekend-weekday)/weekday) of six pollutant concentrations in various functional zones nationwide. *$P < 0.05$.

### 3.3 Spatial variation

### 3.3.1 Six geographical regions

In North and Northwest China, the concentrations of $PM_{2.5}$, $PM_{10}$, $SO_2$, $NO_2$, and CO are notably high (as shown in Figure 6). Specifically, the concentrations of $PM_{2.5}$ and $PM_{10}$ are 8.07 μg/m³ and 28.2 μg/m³ higher, respectively, than in other regions. This disparity can be attributed to North and Northwest China being the epicenter of China's traditional industrial base, endowed with abundant coal and mineral resources. The presence of numerous large-scale enterprises, including coal-fired power plants, the steel industry, and the non-ferrous metal industry, contributes to significant pollutant emissions. Furthermore, the reliance on coal combustion for heating during the winter months exacerbates pollutant emissions (Wang et al., 2014; Li et al., 2019b). In contrast, the eastern and northern parts of China exhibit elevated levels of $O_3$, with concentrations 6.84 μg/m³ higher than those in other specified regions. The dense population and thriving industry in these areas result in a substantial emission of ozone precursors, particularly during the warmer seasons, the higher temperature in East and North China further promotes the photochemical reaction of $O_3$ (He et al., 2023b).

In North China, Northeast China, and East China, the concentration of various pollutants in transportation zone is notably higher than in other functional zones, while the concentration of various pollutants in public management and service zone is the lowest. This is attributed to the well-developed transportation networks, ongoing expansion of transportation infrastructure, and substantial vehicle ownership in these regions, which collectively contribute to elevated emissions of road traffic pollutants (Guo et al., 2022). In South-Central China, the concentrations of pollutants in commercial and industrial zones are comparatively higher, while transportation zone exhibits the lowest levels. For instance, the levels of $PM_{2.5}$ and $NO_2$ in transportation zone are respectively 4.75 and 5.00 μg/m³ lower than in other functional zones. This may be attributed to the larger and more dispersed areas of transportation zone in central and southern China, which often feature extensive greening and open spaces, which facilitate the dilution and dispersion of pollutants, leading to more dispersed traffic pollution (Hong

and Jin, 2021; Magazzino and Mele, 2021). The O₃ concentrations in residential and transportation zones in southwest China are the lowest, with a notable 8.42 μg/m³ reduction compared to other functional zones. In contrast, the concentrations of other

pollutants in these two functional zones are the highest. Furthermore, in terms of industrial zone, the concentrations of various pollutants in Southwest China are significantly lower than those in other regions ($P < 0.05$). However, no clear pattern of pollutant concentration is observed across different functional zones in Northwest China.

According to the pollution situation of the above functional zones, the focus of air pollution control in different regions is different. In North, Northeast, and East China, the primary focus of air quality management should be on transportation-related

emissions. For Central and Southern China, the emphasis should shift towards catering services and industrial emissions. The Southwest region requires attention to both residential and transportation-related pollution sources. In contrast, Northwest China necessitates a comprehensive approach to air quality management, addressing multiple pollution sources simultaneously.

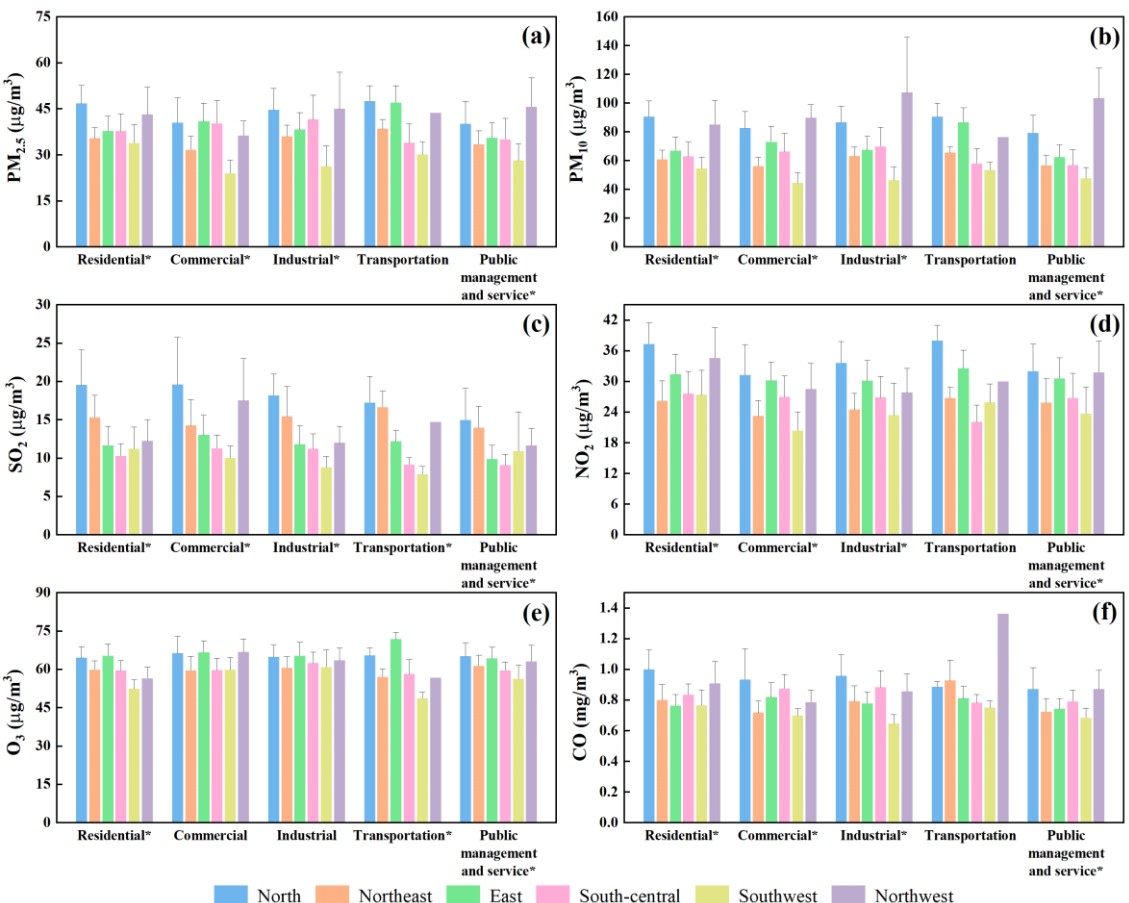

**Figure 6.** Concentrations of PM$_{2.5}$ (a), PM$_{10}$ (b), SO$_2$ (c), NO$_2$ (d), O$_3$ (e), and CO (f) in each functional zone of the six geographical regions. *$P < 0.05$.

### 3.3.2 Six urban agglomerations

The Beijing-Tianjin-Hebei urban agglomeration (BTH) exhibits a pronounced high concentration of pollutants, as shown in

Figure 7. BTH, recognized as the region with the highest pollutant emission intensity in China, experiences a climate characterized by static and stable weather conditions, weak winds, and a relatively low boundary layer height, which collectively provide favorable atmospheric conditions conducive to the formation and accumulation of pollutants (He et al., 2020; Zhang and Cao, 2015). The Greater Bay Area (GBA) demonstrates significantly lower concentrations of PM$_{2.5}$, PM$_{10}$, and CO compared to the other urban agglomerations. Its proximity to the ocean and advantageous geographical positioning

facilitate the dispersion of pollutants, resulting in a comparatively lower pollution level (Shen et al., 2019). The Yangtze River Delta urban agglomeration (YRD) records the highest concentration of O$_3$, exceeding 64.2 μg/m³, while the Chengdu-

Chongqing urban agglomeration (CC) shows the lowest, with levels below 52.7 μg/m³. These findings corroborate the results of Zhao et al.(Zhao et al., 2021b), highlighting the correlation between high population density and anthropogenic emissions of $O_3$ precursors in the YRD. The Northern Slope of Tianshan Mountain urban agglomeration (NSTM) is marked by elevated levels of particulate matter. Human activities and dust events contribute to the significant production of $PM_{2.5}$ and $PM_{10}$, with the high concentration of these particles being influenced by wind direction and speed (Luo et al., 2023).

For BTH and YRD, the concentrations of pollutants in industrial and transportation zones are notably higher, particularly the $NO_2$ levels in transportation zone, which significantly surpass those in other functional zones, which are 40.5 μg/m³ and 37.0 μg/m³, respectively. This disparity is primarily attributed to the economic prosperity, dense population, and high-density traffic flow in these regions, with vehicle emissions identified as the predominant source of $NO_2$ (Yang et al., 2018). Furthermore, in terms of industrial zone, the $SO_2$ concentration of BTH obviously surpasses that of other urban agglomerations ($P < 0.01$), reaching 15.6 μg/m³. The industrial structure of BTH is dominated by heavy industry, with industrial land use exhibiting a high degree of concentration, such as steel, chemical industry, building materials, etc. These sectors serve as the primary contributors to $SO_2$ emissions. In the case of TC and CC, elevated pollutant levels are observed in commercial and industrial zones. The thriving catering industry in these regions, combined with the concentrated distribution of commercial and industrial zones resulting from topographical and planning constraints, contributes to a high emission intensity of pollutants per unit area (Liu et al., 2020). From the perspective of transportation zone, the $NO_2$ concentration in TC is significantly lower than that in other urban agglomerations ($P < 0.01$), standing at merely 21.2 μg/m³. This is attributed to the open terrain of TC's transportation zone, which facilitates the dispersion of pollutants. The GBA exhibits the lowest $O_3$ concentration in commercial zone, measured at 53.1 μg/m³, while other pollutants are highest in commercial zone. The higher concentration of various pollutants in NSTM's residential zone can be ascribed to the substantial use of loose coal burning from 2017 to 2022, resulting in substantial pollutant emissions.

In light of the varying pollution profiles within the aforementioned functional zones, the focus of atmospheric pollution mitigation in different urban agglomerations is different. BTH and YRD should focus on industrial and transportation zones. TC requires attention to commercial and industrial zones. Meanwhile, GBA should prioritize the mitigation of pollution on commercial zone.

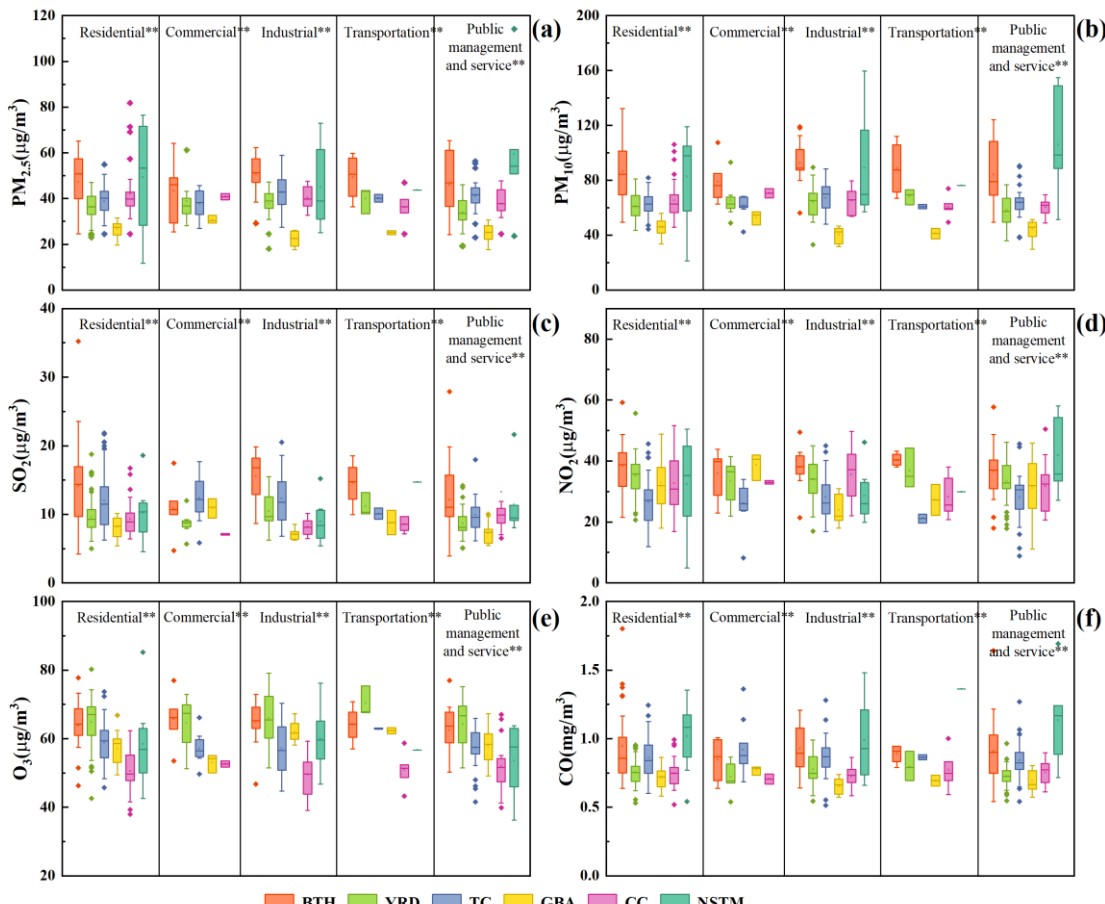

**Figure 7.** Comparisons in concentrations of $PM_{2.5}$ (a), $PM_{10}$ (b), $SO_2$ (c), $NO_2$ (d), $O_3$ (e), and CO (f) in each functional zone of the six urban agglomerations. BTH: Beijing-Tianjin-Hebei urban agglomeration; YRD: Yangtze River Delta urban agglomeration; TC: Triangle of Central China; GBA: Greater Bay Area urban agglomeration; CC: Chengdu-Chongqing urban agglomeration; NSTM: Northern Slope of Tianshan Mountains urban agglomeration. **$P < 0.01$.

## 3.4 Analysis of influencing factors

### 3.4.1 The impact of governance measures

In 2018, the Chinese government launched the Three-Year Action Plan for Winning the Blue Sky Defense Battle, which identified three key areas for air pollution control: the Beijing-Tianjin-Hebei region and its surrounding areas, the Yangtze River Delta region, and the Fen-Wei Plain (Figure S6). The Beijing-Tianjin-Hebei region and its surrounding areas include 28 cities, such as Beijing; the Yangtze River Delta region includes 41 cities, such as Shanghai; the Fen-Wei Plain includes 11 cities, such as Xi'an. Figure 8 shows the particulate matter in the Beijing-Tianjin-Hebei region and its surrounding areas has seen the greatest improvement, with $PM_{2.5}$ and $PM_{10}$ concentrations decreasing by 10.5% and 9.8%, respectively. The improvement in $SO_2$ and CO in the Beijing-Tianjin-Hebei region and the Fen-Wei Plain is the most significant, with the $SO_2$ and CO concentrations in the Beijing-Tianjin-Hebei region falling by 23.8% and 13.3%, respectively, and in the Fen-Wei Plain by 30.6% and 15.2%, respectively. The $NO_2$ levels in key areas are slightly better than those in non-key areas. These results indicate that since the implementation of the Three-Year Action Plan for Winning the Blue Sky Defense Battle, the air pollution control in key areas has achieved significant results.

The Beijing-Tianjin-Hebei region and its surrounding areas demonstrated the most significant improvement in transportation zone for particulate matter ($PM_{2.5}$ and $PM_{10}$), with a reduction rate of 12.68% and 11.45%, respectively. However, commercial zone showed the least improvement, with a reduction rate of 9.28% for $PM_{2.5}$ and 8.34% for $PM_{10}$. This indicates that traffic areas in this region have achieved remarkable dust control. The improvement of transportation zone for gaseous pollutants ($SO_2$, $NO_2$, and CO) was the least significant, with decline rates of 16.72%, 7.08%, and 11.81%, respectively, suggesting that

efforts to control automobile exhaust should be intensified in this region. In contrast, the Yangtze River Delta region saw the least improvement in particulate matter in transportation zone, while gaseous pollutants showed the greater improvement, indicating a need to strengthen dust control in this region's transportation zone. The industrial zone in Fen-Wei Plain has the

smallest improvement in the concentration of various pollutants, because it has abundant coal reserves and intensive energy industries (such as coal-fired power plants, iron and steel smelting, coal coking, etc.), resulting in more industrial pollutants and difficult improvement (Bai et al., 2021). The improvement in residential and commercial zone in other area is the greatest, while transportation zone has seen the least progress. It is clear that the Beijing-Tianjin-Hebei region and its surrounding areas, the Yangtze River Delta, and other area all need to enhance air pollution control in their transportation zone.

The rebound range of $O_3$ in three key areas is significantly higher than that in other area. The rebound rate of ozone in residential and public management and service zone is relatively large. The 1.9% increase in the $O_3$ rebound rate in residential zone in the Fen-Wei Plain is particularly noteworthy. This increase in ozone can be attributed to the elevated levels of active oxygen-containing volatile organic compounds (OVOCs), especially the ongoing rise in formaldehyde (Lin et al., 2023). The aforementioned results indicate the necessity of promoting collaborative governance for $PM_{2.5}$ and $O_3$.


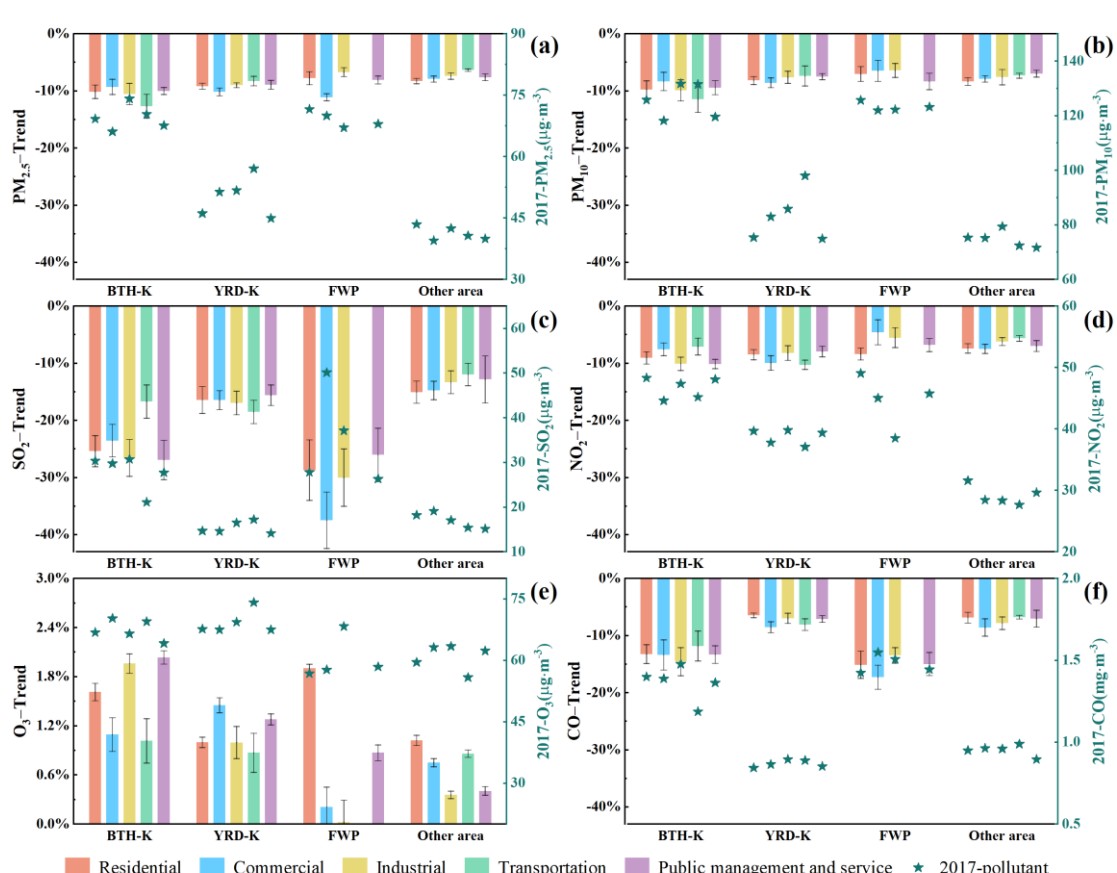

**Figure 8.** Annual variation trend of $PM_{2.5}$ (a), $PM_{10}$ (b), $SO_2$ (c), $NO_2$ (d), $O_3$ (e), and CO (f) concentrations in various functional zones of the three key areas and other area. BTH-K: Beijing-Tianjin-Hebei region and its surrounding areas; YRD-K: Yangtze River Delta region; FWP: Fen-Wei Plain. (The left y-axis presents the temporal variation in pollutant concentrations from 2017
to 2022, the right y-axis displays the pollutant concentrations in 2017.)

### 3.4.2 The impact of altitude

Figure 9 shows the average concentrations of $PM_{2.5}$, $PM_{10}$, $SO_2$, $NO_2$, and CO during 2017-2022 exhibit a general decrease with increasing altitude due to the predominant concentration of atmospheric pollution sources in China's lower eastern region.
Additionally, within the boundary layer, convection movement lifts pollutant particles upward, leading to their dilution (Chen et al., 2024; Rohde and Muller, 2015). Conversely, the average concentration of $O_3$ increases with elevation and demonstrates

a consistent relationship with altitude as previously observed by Ma et al. (2021). The reduction in $SO_2$ and CO concentrations at altitudes between 500 and 1000 m was notably greater than that at other altitudes. Similarly, improvements in $PM_{10}$ and $NO_2$ concentrations above 2000 m were significantly greater compared to lower altitudes.

The reduction in particulate matter and $NO_2$ is most significant at high altitudes (above 2000 m), with diminishing improvements observed between 0-2000 m as altitude increases. Meanwhile, the improvement of $SO_2$ and CO decreases with the increase of altitude above 500 m. These phenomena are primarily influenced by a combination of factors, including emission sources, dilution effects, meteorological conditions, and chemical reactions. In high-altitude areas, meteorological conditions such as wind speed and turbulence are conducive to the dispersion and dilution of pollutants, thereby promoting

the improvement of particulate matter and $NO_2$ concentrations (Román-Cascón et al., 2023). Additionally, $SO_2$ and CO in the atmosphere can be chemically transformed into other pollutants, such as sulfate aerosols and organic aerosols. In higher-altitude regions, the rates and extents of these chemical reactions may change, affecting the degree of improvement in $SO_2$ and CO concentrations (Quan et al., 2021).

In low-altitude regions (below 1000 m), the reduction of these pollutants in transportation zone was minimal. Conversely, in

high-altitude regions (above 1500 m), a significant decrease in these pollutants was observed in both residential and industrial zones. This may be due to the higher density of vehicular traffic in lower altitude areas, coupled with reduced human activity in higher altitude regions. The low-altitude regions are characterized by dense populations and substantial traffic flow, which lead to concentrated and high-intensity vehicle exhaust emissions. Consequently, a substantial number of pollutants, including particulate matter, nitrogen oxides, and volatile organic compounds, are continuously released into the atmosphere. This results

in a high baseline concentration of pollutants, making it challenging to achieve significant improvements in air quality (Lopez-Aparicio et al., 2025). In contrast, high-altitude areas have limited scales and quantities of residential and industrial land. As a result, the total volume of pollutants emitted from human activities is relatively small. Therefore, under the favorable influence of natural conditions, the concentration of pollutants in these regions is more amenable to substantial improvement.

$O_3$ rebounds more with the increase of altitude, especially in the high-altitude area above 2000 m. The possible reason is that

global warming leads to a significant increase in the height of the atmospheric boundary layer, which promotes the transmission of $O_3$ from the upper air to the surface, resulting in a substantial rebound of $O_3$ (Liu et al., 2021). The higher the altitude, the more obvious the rebound will be. In particular, the $O_3$ concentration in residential zone has the largest rebound, up to 4.85%, which needs to be paid attention to.

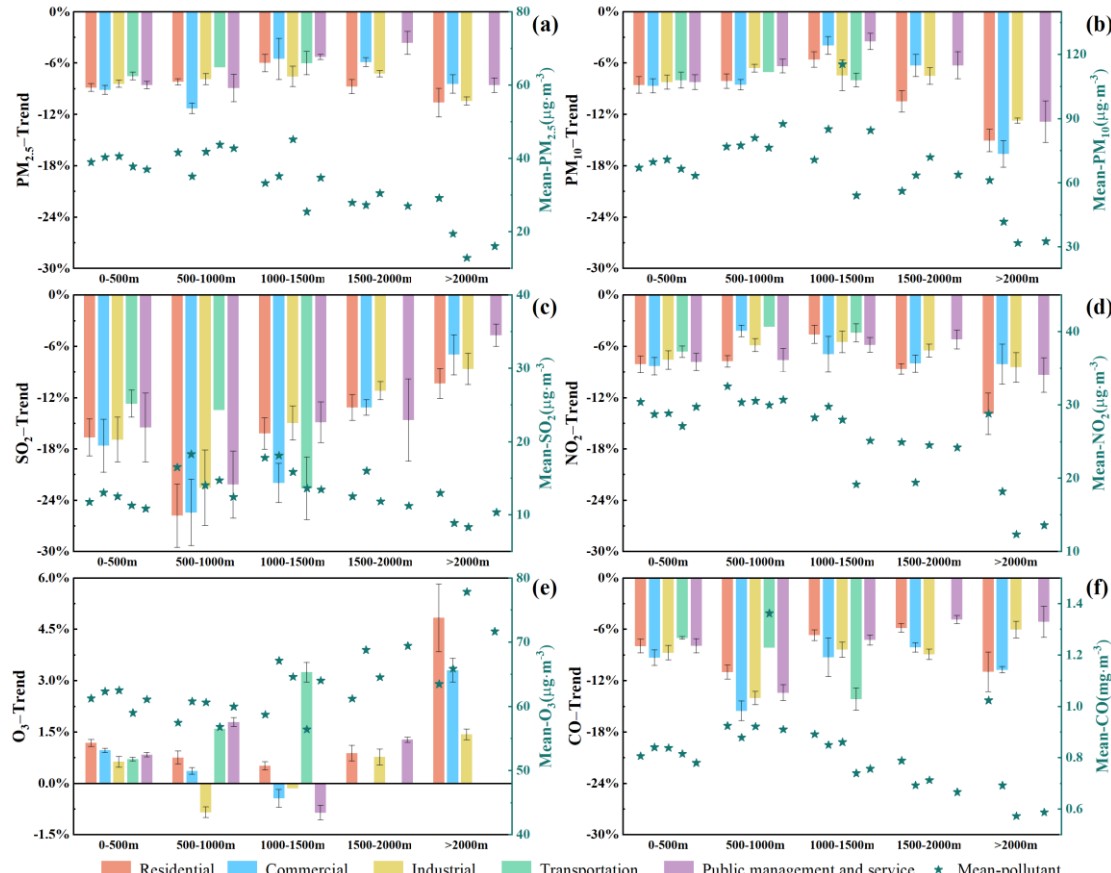


**Figure 9.** Annual variation trend of $PM_{2.5}$ (a), $PM_{10}$ (b), $SO_2$ (c), $NO_2$ (d), $O_3$ (e), and CO (f) concentrations in various functional zones at different altitudes. (The left y-axis presents the temporal variation in pollutant concentrations from 2017 to 2022, the right y-axis displays the mean pollutant concentrations over the same period.)

### 495 3.4.3 The impact of urban scale

The urban scale was categorized according to population density, and comparisons were made between high-density and low-density cities regarding the variations in pollutant concentrations across different functional zones (Figure 10). Specifically, cities with a population density exceeding 510 persons/km² (the 70th percentile) were defined as high-density, whereas those with a density below 151 persons/km² (the 30th percentile) were considered low-density.

The results indicate that cities with high population density exhibit significantly higher concentrations of pollutants compared to those with low population density. In cities with low population density, the improvement in particulate matter ($PM_{2.5}$ and $PM_{10}$) levels in public management and service zone is minimal. In addition, the reduction of $SO_2$, $NO_2$, and CO is more pronounced in residential and commercial zones, while the improvement in transportation zone is the least significant. In cities with high population density, the improvement in $PM_{2.5}$ and $SO_2$ concentrations in transportation zone is notably smaller

compared to other functional zones, at 8.05% and 14.2%, respectively. This can be attributed to the relatively fixed sources of pollutant emissions in transportation zone, particularly in high population density urban areas where traffic flow remains challenging to significantly reduce even with optimized traffic management, thereby posing greater difficulty in achieving improvements (Lopez-Aparicio et al., 2025). Additionally, the high proportion of impervious pavement in transportation zone contributes to the "heat island effect," which hinders the dispersion of pollutants (Yuan et al., 2018). Therefore, future efforts

should prioritize pollution control in transportation zone.

From the perspective of different functional zones, the differences of variation in pollutant levels among urban regions with high population density is less pronounced compared to those with low-density (with the exception of $SO_2$ and $O_3$). When examining the same functional area across cities of varying population densities, significant differences are observed in public management and service zone, where $PM_{2.5}$ and $PM_{10}$ levels differ by 4.5% and 3.8%, respectively. In contrast, residential

zone exhibits minimal variation, with PM$_{2.5}$ and PM$_{10}$ levels differing by only 0.2% and 1.0% (Table S1).

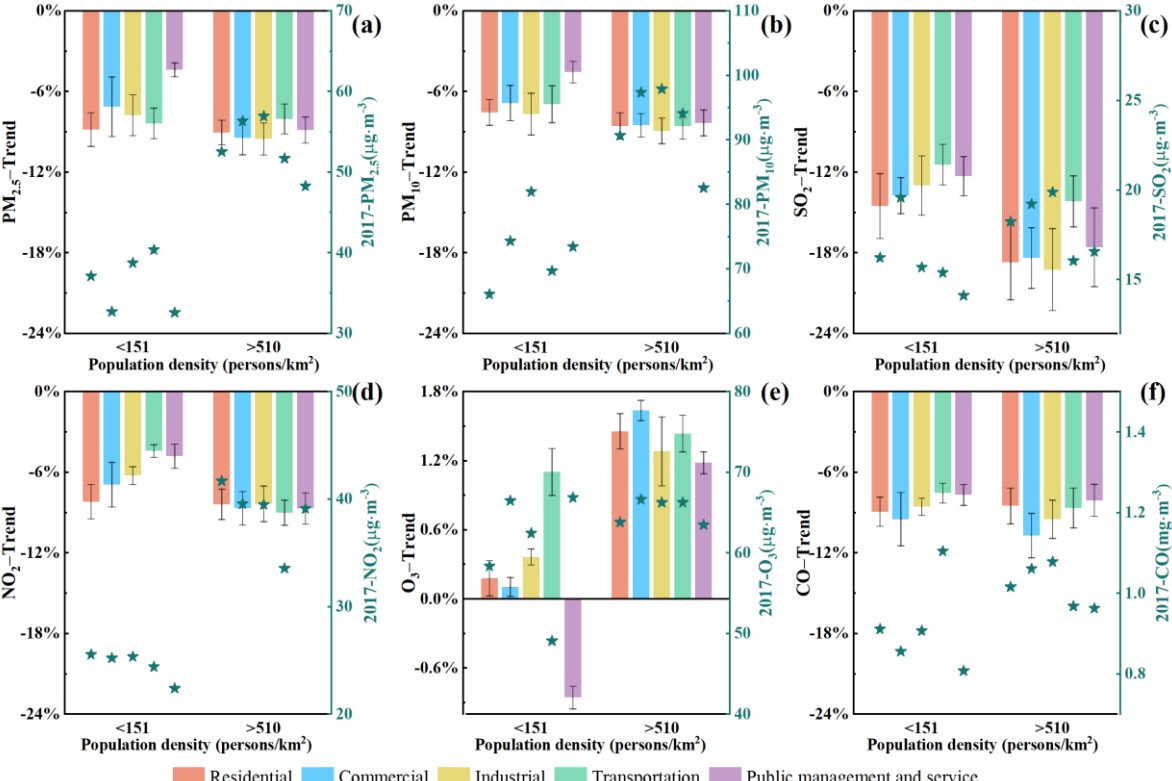

**Figure 10.** Annual variation trend of PM$_{2.5}$ (a), PM$_{10}$ (b), SO$_2$ (c), NO$_2$ (d), O$_3$ (e), and CO (f) concentrations in various functional zones in different urban scale. (The left y-axis presents the temporal variation in pollutant concentrations from 2017 to 2022, the right y-axis displays the pollutant concentrations in 2017.)


### 3.4.4 Policy implications

(1) Strengthen the control of air pollution in transportation zone.

The reduction rate of pollutant concentration in transportation zone was considerably lower than that in other regions. This indicates that more stringent air pollution control measures are imperative in transportation-related areas. Regarding vehicle
exhaust emission control, it is essential to rigorously enforce vehicle exhaust emission standards. A comprehensive supervision system must be established with reinforced oversight throughout the vehicle exhaust testing process to guarantee that every on-road vehicle consistently complies with emission standards. For vehicles with excessive exhaust emissions, a mandatory maintenance or elimination mechanism should be implemented. This approach aims to reduce exhaust pollutant emissions from motor vehicles at the source and enhance air quality. In addition, to foster the sustainable development of urban
transportation, it is necessary to encourage and support the development and extensive application of new energy vehicles (NEVs).

(2) Strengthen the supervision and management of emissions in commercial zone.

The phenomenon of the "weekend effect" of sulfur dioxide in commercial zone serves as a spotlight on the significant influence that emissions from the catering industry exert on air quality. In light of this, it is imperative for the relevant authorities to
intensify environmental oversight of catering enterprises. This includes the rigorous enforcement of emission standards, mandating the installation of high-efficiency grease fume purification and waste gas treatment equipment in these enterprises, as well as ensuring the regular maintenance and monitoring of such equipment. Furthermore, in response to the emission peak triggered by the surge in weekend activities within commercial areas, the formulation of a staggered operation policy presents itself as a viable solution. By strategically scheduling the timing of commercial activities, it is possible to prevent the
concentrated release of pollutants. Meanwhile, commercial enterprises should be encouraged to embrace more environmentally friendly business operation models.

(3) Formulate differentiated urban air pollution control strategies.

Significant temporal and spatial variations exist in air pollutant concentrations across different urban functional zones. Therefore, future air pollution control initiatives should integrate air pollution control strategies, address the pollution issues of each functional zone in a coordinated manner, and strengthen cross-regional cooperation and joint prevention efforts. Given the distinct characteristics of individual functional zones, it is essential to develop and execute emission reduction strategies with precision, informed by spatio-temporal differentiation to continuously enhance the process of air quality improvement.

## 4 Conclusion

Drawing on air quality observation data from 336 Chinese cities spanning the years 2017 to 2022, this study conducts a comprehensive analysis of the spatiotemporal evolution characteristics and potential influencing factors of air pollutants in various urban functional zones. The key findings are outlined below.

In terms of time scale, our analysis reveals a general downward trend in the concentrations of key air pollutants, especially in residential, commercial, and industrial zones, which can be largely attributed to the implementation of stringent national pollution control measures. The concentration of $SO_2$ decreased by more than 50%, representing the most significant reduction among all pollutants. This reduction is indicative of the effectiveness of these policies in combating air pollution. The power and industrial sectors have registered significant achievements in reducing emissions and have taken a leading role in driving phenomenal improvements in air quality. The rate of decrease in pollutant concentration in transportation zone was significantly lower than in other zones, with $NO_2$ showing the least reduction at 28.3%. However, ozone levels rebound notably in densely populated residential and commercial zones. Seasonal variation indicates a general downward trend in the fluctuation of these pollutants, with industrial zone showing the most pronounced seasonal variations and public management and service zone the least. Diurnal variation analysis reveals that commercial, industrial, and transportation zones experience the greatest daily fluctuations, in contrast to public management and service zone, which show minimal variation. The "weekend effect" for pollutants varies by functional zones. For particulate matter, a strong "weekend effect" is seen in residential and transportation zones due to increased human activities in these zones. $SO_2$ shows a notably "weekend effect" in commercial zone, likely due to increased emissions from catering. For commercial zones, hubs of economic transactions and social activities, exhibit a distinct "weekend effect," with increased emissions due to heightened activities. This finding calls for effective management of commercial operations, especially during peak times, and the potential for greener business practices.

Spatial analysis revealed significant differences in pollutant concentrations in different regions and different functional zones. In North China, Northeast China, and East China, the concentration of various pollutants in transportation zone is higher than that in other functional zones. The concentration of pollutants in commercial and industrial zones is higher in South-Central China. The concentration of $O_3$ in residential and transportation zones in southwest China is the lowest, while the concentration of other pollutants in these two functional zones is the highest. Beijing-Tianjin-Hebei city cluster and Yangtze River Delta city cluster are particularly prominent in the concentration of pollutants in industrial zone and transportation zone. Notably, $NO_2$ levels in transportation zone of BTH and YRD substantially exceed those in other functional zones, with concentrations reaching 40.5 μg/m³ and 37.0 μg/m³, respectively. The concentrations of pollutants in commercial zone and industrial zone in the middle reaches of Triangle of Central China and Chengdu-Chongqing city cluster are also relatively elevated.

The study also found that treatment measures, altitude, and urban scale have an impact on the concentration of pollutants in different functional zones. Overall, the improvement of pollutants in key areas implementing stricter air pollution control measures is greater than that in other areas. Nevertheless, it is imperative for all areas to enhance air pollution control in their transportation zone. The rebound rate of $O_3$ in key areas is significantly greater than that in other area, especially in residential zone and public management and service zone. The influence of altitude cannot be ignored. In low-altitude regions (below

1000 m), the reduction of these pollutants in transportation zone was minimal. Conversely, in high-altitude regions (above 1500 m), a significant decrease in these pollutants was observed in both residential and industrial zones. The effect of urban scale on pollutant improvement ranges is as follows: in cities with high population density, the differences in the extent of pollutant improvement among functional zones are relatively small. Conversely, in cities with low population density, these differences are more pronounced. Additionally, urban scale has the most significant impact on public management and service zone and the least impact on residential zone.

To sum up, the intricate link between urbanization and air quality, highlighting the need for continuous monitoring and the development of zone-specific air quality strategies. The findings advocate for adaptive urban planning that takes into account the unique challenges posed by urban functional zones and the necessity for innovative pollution mitigation approaches. In essence, the research contributes to a deeper understanding of the complex dynamics of air quality in urban China. It offers valuable guidance for policymakers and urban planners in crafting effective and targeted air quality management strategies, which are essential for achieving sustainable urban environments. The insights gained from this study are not only pertinent to China but also provide a framework for understanding urban air quality challenges and developing appropriate responses in other urban areas globally.

However, this study also possesses certain limitations that warrant acknowledgment. The spatial distribution of monitoring stations differs across regions, which may result in a potential loss of variability when data are grouped regionally. Furthermore, this study has only analyzed the influence of several individual factors without conducting a comprehensive evaluation of their relative contributions. In future research, as simulation technologies continue to advance, we will strive to conduct a more thorough analysis of these integrated effects. Additionally, we plan to actively undertake in-depth case studies on representative cities to provide enhanced guidance for urban planning and management.

**Data availability.** All data needed to evaluate the conclusions in the paper are present in the paper and/or the Supplement. Also, all data used in the study are available from the corresponding author upon request.

**Supplement.** The supplement related to this article is available online at:

**Author contributions.** All authors made substantial contributions to this work. **Lulu Yuan:** Methodology, Visualization, Formal analysis, Resources, Data curation, Writing – original draft. **Wenchao Han:** Conceptualization, Methodology, Visualization, Resources, Writing – review & editing, Supervision. **Jiachen Meng:** Visualization, Data curation, Writing – review & editing. **Yang Wang:** Visualization, Writing – review & editing, Supervision. **Haojie Yu and Wenze Li:** Data curation, Writing – review & editing.

**Competing interests.** The authors declare that they have no conflict of interest.

**Acknowledgements.** This work was supported by the National Key Research and Development Program of China (2022YFC3703004) and the National Natural Science Foundation of China (42301093).

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
