# Peer review of "Uncovering the Impact of Urban Functional Zones on Air Quality in China"

_EGUsphere, 2024_

## Referee Comment (RC2)

**Introduction**

This manuscript evaluates air pollutant trends in time (annual, seasonal, daily) and space (regions, agglomerations) from 1364 monitoring sites across China which have been individually categorized into one of five distinct urban functional zones. The authors have sliced, diced and presented the data in many different ways and go on to discuss the results in relation to social (management) and environmental factors.

**Overall Quality of Pre-print**

The authors have undeniably acquired, analysed and visualised large volumes of monitoring data (hourly data for 1364 sites x 6 years) but the manuscript is highly descriptive, not particularly analytical in nature and provides little novelty in approach or outcome in its current form. Some elements of the paper also assume that the reader has a good knowledge of China which is not necessarily the case.

The introduction suggests that other researchers have already commented on reductions in air pollutants across China, with regional differences, with the possible exception of ozone, so the novel element here is analysis of trends in time and space through the lens of urban functional zones. I think the emphasis on urban functional zones is interesting because these provide a different means of grouping the data and establishing trends by sector which may ultimately be related to the success or otherwise of management actions. The danger, of course, is that by grouping data everything collapses to the mean, and variation is lost. For example, subtleties linked to differences in the scale and form or urban areas, or local and regional climatic influences cannot be considered.

This manuscript, then, demonstrates how ambient pollutant concentrations have reduced over space and time. Interestingly, it does not include any information on emissions. It would be good to include some information on emissions in a revised version of this manuscript – the authors could then evaluate on which emission reduction strategies (and in which sectors) had been most successful in reducing ambient concentrations.

I would particularly like to see better quantification of differences in pollutant concentrations in space and time for air pollutants in the different urban functional zones. It should be possible, for example, to test whether differences between mean concentrations across zones are statistically significant. Likewise, it should be possible to compute trend lines over individual time series and report rates of change and significance of change. The authors could then discuss their main results with more confidence, and include some headline statistics in the abstract.

The manuscript at present does not have a discussion section. I think that once the results have been re-analysed, with more statistical rigour, the main results and discussion points should become more apparent. Using urban functional zones, the

authors should be well-placed to comment on the significant progress that has been made towards reducing harmful ambient pollutant concentrations, with more rapid progress being made in some parts of the country than others presumably attributed to a combination of more effective emission reduction strategies target at specific source sectors and favourable environmental conditions.

It would also be useful for the authors to reflect on the limitations of their approach, for example, loss of variation caused by grouping data into zones, unequal numbers of sites per zone, broader issues of scale and local topographic influences.   I would personally place less emphasise on ozone in the revised manuscript – because regional influences are likely to far outweigh local influences and differences between urban functional zones are likely to be negligible.   It is also less responsive to control measures (as your analysis demonstrates).

**Specific Comments**

Section 2.  Subsection 2.1.  Please improve the description of the study area.  Could you include the 6 major regions as polygon outlines and the 6 major urban agglomerations as colour filled polygons on Figure 1?  Could you show the 336-prefecture cities as points on Figure 1.  These changes would provide the reader with a much clearer understanding of the different geographical units used in this study.

Section 2. Subsection 2.2.  Please check your weblinks and ensure they are all accessible to the wider scientific community.  I could not reach:

- beijingair.sinaapp.com               (pollutant data)
- https://data.ess.tsinghua.edu.cn/   (global land cover data)

The latter redirected me to another site, but this hung.

Section 2.  Subsection 2.3.2. Please could you provide a clearer explanation of the way in which you used GIS techniques to categorize each monitoring station by urban functional zone.  I presume you conducted a point-in-polygon test?  Did you consider constructing a buffer and assigning the monitoring station to the dominant functional zone within that buffer (to reduce edge effects)?

Section 3. Subsection 3.1. Please could you provide a new table summarizing the number of monitoring stations per urban functional zone.  You could also summarize the number of monitoring stations per urban functional zone per region, and per urban agglomeration.  This would be far more effective than the map (Figure 2) which could be deleted.

Section 3.  Subsection 3.2.1.  Could you come up with a more effective way of showing reductions in pollutant concentrations over time?  Your left hand plots show percentage reduction over time and presents all pollutants on similar sized plots but with very

different scales on the y-axes (ranging from -8% for $O_3$ to -60% for $SO_2$). Could you either plot these all on the same vertical percentage scale, or, better still, plot on an absolute scale (micrograms per cubic metre) with trend lines. These could usefully a) quantify absolute reductions in pollutant concentrations over time and b) determine whether reductions in some urban functional zones were greater than in others (through an analysis of gradients) which could link to policy measures. Some of this new quantitative information could be included in a revised abstract. Your right hand plots are more difficult to interpret. Are they necessary?

Section 3. Subsection 3.2.2. (Seasonal). Figure 4 highlights seasonal variations in $PM_{2.5}$ and $O_3$ across the 5 urban functional zones over a 6 year period. I have two comments to make here. Firstly, you only show 2 pollutants, when others, such as $SO_2$ presumably exhibit significant seasonal variation. Why not include these too? Secondly, despite having a separate plot for each urban functional zone, it is difficult to establish whether there are meaningful differences between these zones. Are they all behaving the same, or are some behaving differently. Could you test this, statistically?

Section 3. Subsection 3.2.2 (Daily). Figure 5 shows daily variations in pollutant concentrations per urban functional zone and you conclude that some pollutant concentrations vary more in some urban functional zones than others. Could you be more quantitative in your assessment here, please? Could you statistically test for differences? For example, $PM_{2.5}$ looks to exhibit much less diurnal variation in the public management functional zone than other functional zones, but is this statistically significant?

Section 3. Subsection 3.3.1. Again, I wonder if there are more effective ways of communicating differences over geographical regions and whether these are statistically significant. Could, for example, the bar charts be organised by urban functional zone rather than by geographical region, so we have 6 adjacent bars showing pollutant concentrations for the residential, commercial, industrial, transport and management zones side-by-side? You could label these N, NE, E, SE, SW and NW, respectively. Could these also be tested to determine a) whether concentrations in one zone were significantly different to those in another zone and b) whether concentrations for a single zone, e.g., industry, were significantly different in different geographic regions? You do something similar in Figure 7 which works quite well.

Section 3. Subsection 3.3.1. Figure 7 is more effective than Figure 6 and focusses on agglomeration zones not regions. Again, it would help the discussion (management of pollution sources) if differences in pollutant concentrations in different functional zones in different urban agglomerations could be quantified in some way. For example, are $SO_2$ concentrations from industrial sources in BTH and TC significantly different to those in YRD, GBA, CC and NSTM?

Section 3. Subsection 3.4.1. It is good that you link improvements in air quality to management actions and here you talk more specifically about 3 key regions which seem different (larger) to those included on Figure 1. I wonder, do we need additional figures at this stage, or could content from line 365 onwards form part of the discussion.

Section 3. Subsection 3.4.2. I wondered to what extent you were testing altitudinal effects and to what extent you were testing urbanisation effects – with highly populated urban areas and associated infrastructure generally located at lower altitudes. Please check your interpretation of this section.

Section 3. Subsection 3.4.3. I wondered whether this material would be better placed earlier in the manuscript, maybe after the material on seasonal variations, since this is essentially a story of pollution by time? I also wanted to challenge your meanings of positive and negative weekend effects. For me, less pollution is a positive thing, not a negative thing. Here, I think you interpret 'positive weekend effect' as concentrations being higher than in the week, and vice-versa for 'negative weekend effect'. Again, it would be useful to test whether weekday and weekend differences in pollutant concentrations between different urban functional zones were statistically significant.

Section 4. You do not appear to include a discussion section in your paper. I would restructure your manuscript such that you have a substantive discussion section that reflects on the main findings of your analysis, comparing and contrasting your results to those reported by others. I would particularly focus upon the novel element of your study, the use of urban functional zones, and the differential impacts of air quality management on different pollutants in different zones in different regions. I would also comment on the limitations of your study – for example, you do not really consider scale effects or the issue of variable sample size, with some of your results based on many monitoring sites per urban functional zone and others very few. There is also the issue as to whether you should classify a point based on its immediate interaction with the land use, or some broader geographical zone (e.g., 1km buffer, 5km buffer).

Section 5. I would encourage you to revisit your conclusions once you have revised your analysis taking a more critical, quantitative approach to analysing spatial and temporal trends across your different urban functional zones. I do like the method you propose here, and think there is much to be learned from evaluating data grouped across areas with common geographical characteristics to establish which management actions have been most effective at reducing which pollutants across which sectors.

---

## Author Comment (AC1)

Dear Editor and Reviewers,

Thank you for your comments concerning our manuscript entitled "Uncovering the Impact of Urban Functional Zones on Air Quality in China". Those comments are all valuable and very helpful for revising and improving our paper, as well as the important guiding significance to our researches. We have studied comments carefully and have made correction which we hope meet with approval. Revisions are indicated within the text using track changes. **All of the responses have been addressed in blue and revisions have been addressed in red as following file of response letter**, and page numbers and line numbers in the response are based on the clean version. In the following, we include a point-by-point response to the comments.

Thank you and best regards.

Yours sincerely,
Wenchao HAN & Yang WANG

**Responses to RC2**

**Overall Quality of Preprint:**

The authors have undeniably acquired, analysed and visualised large volumes of monitoring data (hourly data for 1364 sites 6 years) but the manuscript is highly descriptive, not particularly analytical in nature and provides little novelty in approach or outcome in its current form.    Some elements of the paper also assume that the reader has a good knowledge of China which is not necessarily the case.

**Response:**

We sincerely appreciate your constructive comment. The analysis in certain sections of the manuscript could be strengthened. To address this, we have provided additional explanations in the discussion.

"Furthermore, in terms of industrial zone, the $SO_2$ concentration of BTH obviously surpasses that of other urban agglomerations ($P < 0.01$), reaching 15.6 μg/m³. The industrial structure of BTH is dominated by heavy industry, with industrial land use exhibiting a high degree of concentration, such as steel, chemical industry, building materials, etc. These sectors serve as the primary contributors to $SO_2$ emissions." **in Section 3.3.2 Lines 395-399.**

"The reduction in particulate matter and $NO_2$ is most significant at high altitudes (above 2000 m), with diminishing improvements observed between 0-2000 m as altitude increases. Meanwhile, the improvement of $SO_2$ and CO decreases with the increase of altitude above 500 m. These phenomena are primarily influenced by a combination of factors, including emission sources, dilution effects, meteorological conditions, and chemical reactions. In high-altitude areas, meteorological conditions such as wind speed and turbulence are conducive to the dispersion and dilution of pollutants, thereby promoting the improvement of particulate matter and $NO_2$ concentrations (Román-Cascón et al., 2023). Additionally, $SO_2$ and CO in the atmosphere can be chemically transformed into other pollutants, such as sulfate aerosols and organic aerosols. In higher-altitude regions, the rates and extents of these

chemical reactions may change, affecting the degree of improvement in $SO_2$ and CO concentrations (Quan et al., 2021). In low-altitude regions (below 1000 m), the reduction of these pollutants in transportation zone was minimal. Conversely, in high-altitude regions (above 1500 m), a significant decrease in these pollutants was observed in both residential and industrial zones. This may be due to the higher density of vehicular traffic in lower altitude areas, coupled with reduced human activity in higher altitude regions. The low-altitude regions are characterized by dense populations and substantial traffic flow, which lead to concentrated and high-intensity vehicle exhaust emissions. Consequently, a substantial number of pollutants, including particulate matter, nitrogen oxides, and volatile organic compounds, are continuously released into the atmosphere. This results in a high baseline concentration of pollutants, making it challenging to achieve significant improvements in air quality (Lopez-Aparicio et al., 2025). In contrast, high-altitude areas have limited scales and quantities of residential and industrial land. As a result, the total volume of pollutants emitted from human activities is relatively small. Therefore, under the favorable influence of natural conditions, the concentration of pollutants in these regions is more amenable to substantial improvement." **in Section 3.4.2 Lines 464-482**.

Additionally, to enhance readers' understanding of the research context, we have revised Figure 1 accordingly. Specific details can be found in our reply to Comment 1 provided later.

The introduction suggests that other researchers have already commented on reductions in air pollutants across China, with regional differences, with the possible exception of ozone, so the novel element here is analysis of trends in time and space through the lens of urban functional zones. I think the emphasis on urban functional zones is interesting because these provide a different means of grouping the data and establishing trends by sector which may ultimately be related to the success or otherwise of management actions. The danger, of course, is that by grouping data everything collapses to the mean, and variation is lost. For example, subtleties linked to differences in the scale and form or urban areas, or local and regional climatic

influences cannot be considered.

**Response:**

    Thank you for your valuable advice. After thorough deliberation, we acknowledge the issue you raised regarding the loss of differentiation when using the mean of grouped data. In light of this concern and in conjunction with the first reviewer's Comment 9, we have incorporated a new **Section 3.4.3**, which explores the impact of urban scale.

**3.4.3 The impact of urban scale**

The urban scale was categorized according to population density, and comparisons were made between high-density and low-density cities regarding the variations in pollutant concentrations across different functional zones (Figure 10). Specifically, citys with a population density exceeding 510 persons/km² (the 70th percentile) were defined as high-density, whereas those with a density below 151 persons/km² (the 30th percentile) were considered low-density.

The results indicate that cities with high population density exhibit significantly higher concentrations of pollutants compared to those with low population density. In cities with low population density, the improvement in particulate matter ($PM_{2.5}$ and $PM_{10}$) levels in public management and service zone is minimal. In addition, the reduction of $SO_2$, $NO_2$, and CO is more pronounced in residential and commercial zones, while the improvement in transportation zone is the least significant. In cities with high population density, the improvement in $PM_{2.5}$ and $SO_2$ concentrations in transportation zone is notably smaller compared to other functional zones, at 8.0% and 11.4%, respectively. This can be attributed to the relatively fixed sources of pollutant emissions in transportation zone, particularly in high population density urban areas where traffic flow remains challenging to significantly reduce even with optimized traffic management, thereby posing greater difficulty in achieving improvements (Lopez-Aparicio et al., 2025). Additionally, the high proportion of impervious pavement in transportation zone contributes to the "heat island effect," which hinders the dispersion of pollutants (Yuan et al., 2018). Therefore, future efforts should prioritize pollution

control in transportation zone.

From the perspective of different functional zones, the differences of variation in pollutant levels among urban regions with high population density is less pronounced compared to those with low-density (with the exception of $SO_2$ and $O_3$). When examining the same functional area across cities of varying population densities, significant differences are observed in public management and service zone, where $PM_{2.5}$ and $PM_{10}$ levels differ by 4.5% and 3.8%, respectively. In contrast, residential zone exhibits minimal variation, with $PM_{2.5}$ and $PM_{10}$ levels differing by only 0.2% and 1.0% (Table S1).

[Figure]

Figure 10. Annual variation trend of $PM_{2.5}$ (a), $PM_{10}$ (b), $SO_2$ (c), $NO_2$ (d), $O_3$ (e), and CO (f) concentrations in various functional zones in different urban scale.

Table S1. Variation differences in pollutant concentrations among different urban scale (low density-high density).

|  | $PM_{2.5}$ | $PM_{10}$ | $SO_2$ | $NO_2$ | $O_3$ | CO |
|---|---|---|---|---|---|---|
| Residential | 0.23% | 1.01% | 4.17% | 0.19% | -1.28% | -0.43% |
| Commercial | 2.33% | 1.66% | 4.65% | 1.75% | -1.53% | 1.22% |
| Industrial | 1.76% | 1.25% | 6.26% | 2.10% | -0.92% | 0.93% |
| Transportation | -0.33% | 1.63% | 2.73% | 4.59% | -0.33% | 1.10% |
| Public management and service | 4.47% | 3.77% | 5.28% | 3.88% | -2.04% | 0.39% |

**Reference**

Lopez-Aparicio, S., Grythe, H., Drabicki, A., Chwastek, K., Toboła, K., Górska-Niemas, L., Kierpiec, U., Markelj, M., Strużewska, J., Kud, B., and Sousa Santos, G.: Environmental sustainability of urban expansion: Implications for transport emissions, air pollution, and city growth, Environment International, 196, 109310, https://doi.org/10.1016/j.envint.2025.109310, 2025.

Yuan, M., Huang, Y., Shen, H., and Li, T.: Effects of urban form on haze pollution in China: Spatial regression analysis based on PM2.5 remote sensing data, Applied Geography, 98, 215–223, https://doi.org/10.1016/j.apgeog.2018.07.018, 2018.

From a climatic perspective, given that functional zones typically cover very small areas, the climatic conditions within the same region are unlikely to exert differential impacts on distinct functional zones. To comprehensively investigate the influence of climate, we systematically considered various climate zones and meticulously selected research subjects from diverse geographical regions, urban agglomerations, and altitudes for in-depth analysis.

This manuscript, then, demonstrates how ambient pollutant concentrations have reduced over space and time. Interestingly, it does not include any information on emissions. It would be good to include some information on emissions in a revised version of this manuscript – the authors could then evaluate on which emission reduction strategies (and in which sectors) had been most successful in reducing ambient concentrations.

**Response:**

Thank you for your suggestion. At present, the resolution of existing emission inventory data, such as MEIC, is generally at the level of 10 km or higher. However, the distribution of functional zones exhibits a high degree of complexity and diversity (as shown in Figure R2), resulting in a 10×10 km grid encompassing multiple types of functional zones. Due to this complexity, it is challenging to directly apply emission data to the analysis of small-scale functional zones.

[Figure]

Figure R2. Distribution of urban functional zones.

In order to account for the impact of emissions, we have referred to relevant literature. The emissions of $SO_2$ and primary $PM_{2.5}$ have shown a sustained downward trajectory (Figure R3). Notably, the power and industrial sectors have contributed the most to this emission reduction. Consequently, it can be inferred that the reduction in emissions has played a predominant role in uplifting the air quality. We have added the analysis in **Section 3.2.1 Lines 231-233**. "The power and industrial sectors have registered significant achievements in reducing emissions and have taken a leading role in driving phenomenal improvements in air quality (Geng et al., 2024)."

[Figure]

Figure R3. Anthropogenic emissions by sector for 2013–2020 for $SO_2$, $NO_x$, primary $PM_{2.5}$, NMVOC and $NH_3$ (Geng et al., 2024).

**Reference**

Geng, G., Liu, Y., Liu, Y., Liu, S., Cheng, J., Yan, L., Wu, N., Hu, H., Tong, D., Zheng, B., Yin, Z., He, K., and Zhang, Q.: Efficacy of China's clean air actions to tackle $PM_{2.5}$ pollution between 2013 and 2020, Nat. Geosci., 17, 987–994, https://doi.org/10.1038/s41561-024-01540-z, 2024.

I would particularly like to see better quantification of differences in pollutant concentrations in space and time for air pollutants in the different urban functional zones. It should be possible, for example, to test whether differences between mean concentrations across zones are statistically significant. Likewise, it should be possible to compute trend lines over individual time series and report rates of change and significance of change. The authors could then discuss their main results with more confidence, and include some headline statistics in the abstract.

**Response:**

Thanks for your insightful suggestion. We have utilized statistical methods to test the significance of the average concentration differences among different functional zones and carried out trend line fitting for the time series. For the modified and improved contents in this regard, please refer to our responses to the specific comments provided later.

The manuscript at present does not have a discussion section. I think that once the results have been re-analysed, with more statistical rigour, the main results and discussion points should become more apparent. Using urban functional zones, the authors should be well-placed to comment on the significant progress that has been made towards reducing harmful ambient pollutant concentrations, with more rapid progress being made in some parts of the country than others presumably attributed to a combination of more effective emission reduction strategies target at specific source sectors and favourable environmental conditions.

**Response:**

Thanks for your suggestion. Given that some of the results in this article cover a wide range and contain many detailed analyses, after careful consideration, we believe that if a separate discussion section is added, it may make the overall structure of the

article slightly loose. To make it more convenient for readers to read, this paper does not provide a dedicated chapter for discussion. Instead, it integrates result analysis and discussion. While presenting the results, we concurrently carried out relevant discussions. For example, in **Section 3.2.1,** the results are "Significant variations are observed in the reduction of pollutant concentrations across different functional zones. Except for $O_3$, the concentration of pollutants has improved most significantly in residential, commercial, and industrial zones. Compared with 2017, $PM_{2.5}$ concentration had decreased by 34.3%, 35.5%, and 33.8% in these areas by 2022, and $SO_2$ concentration had decreased by 55.6%, 56.4%, and 53.4%, respectively" in **Lines 224-227**; the discussion is "The main reason is that following the implementation of policies such as the "Clean Winter Heating Plan for Northern China (2017–2021)" (NDRC, 2017) and the "Three-Year Action Plan for Winning the Blue Sky Defense Battle", the government intensified control over industrial pollution, promoted clean production in enterprises, implemented clean heating measures in residential areas, and encouraged the use of clean fuels like natural gas and electricity (Song et al., 2023; Wang et al., 2022b). The power and industrial sectors have registered significant achievements in reducing emissions and have taken a leading role in driving phenomenal improvements in air quality (Geng et al., 2024)." in **Lines 227-233**.

We also added additional discussions. "Furthermore, in terms of industrial zone, the $SO_2$ concentration of BTH obviously surpasses that of other urban agglomerations ($P < 0.01$), reaching 15.6 μg/m³. The industrial structure of BTH is dominated by heavy industry, with industrial land use exhibiting a high degree of concentration, such as steel, chemical industry, building materials, etc. These sectors serve as the primary contributors to $SO_2$ emissions."in **Lines 395-399**. "The low-altitude regions are characterized by dense populations and substantial traffic flow, which lead to concentrated and high-intensity vehicle exhaust emissions. Consequently, a substantial number of pollutants, including particulate matter, nitrogen oxides, and volatile organic compounds, are continuously released into the atmosphere. This results in a high baseline concentration of pollutants, making it challenging to achieve significant improvements in air quality (Lopez-Aparicio et al., 2025). In contrast, high-altitude

areas have limited scales and quantities of residential and industrial land. As a result, the total volume of pollutants emitted from human activities is relatively small. Therefore, under the favorable influence of natural conditions, the concentration of pollutants in these regions is more amenable to substantial improvement." in **Lines 476-482**.

Moreover, to present the implications of this paper on management policies in greater detail, we have added **Section 3.4.4**, specifically dedicated to discussing the insights gained from our conclusions for future atmospheric governance policies.

It would also be useful for the authors to reflect on the limitations of their approach, for example, loss of variation caused by grouping data into zones, unequal numbers of sites per zone, broader issues of scale and local topographic influences.    I would personally place less emphasise on ozone in the revised manuscript – because regional influences are likely to far outweigh local influences and differences between urban functional zones are likely to be negligible.    It is also less responsive to control measures (as your analysis demonstrates).

**Response:**

Thanks for your valuable suggestions. In response to your comments, we have incorporated the discussion on the variation loss resulting from data grouping into regions and the imbalance in monitoring site numbers across regions into the limitations, which is located in the conclusion of the revised manuscript. Moreover, we have reduced the emphasis on ozone.

**Specific Comments:**

**Comment 1:** Section 2.    Subsection 2.1.    Please improve the description of the study area.    Could you include the 6 major regions as polygon outlines and the 6 major urban agglomerations as colour filled polygons on Figure 1?    Could you show the 336-prefecture cities as points on Figure 1.    These changes would provide the reader with a much clearer understanding of the different geographical units used in this study.

**Response:**

We sincerely appreciate your constructive suggestion. In response, we have thoroughly revised Figure 1. To enhance clarity and visualization, we have delineated the six major geographical regions of China with distinct polygonal outlines. The six major urban agglomerations are now prominently marked through a color-filling technique. Additionally, all 336 prefecture-level cities are accurately represented by dot symbols.

[Figure]

Figure 1. The location of study area and the distribution map of six geographical regions and six major urban agglomerations. BTH: Beijing-Tianjin-Hebei urban agglomeration; YRD: Yangtze River Delta urban agglomeration; TC: Triangle of Central China; GBA: Greater Bay Area urban agglomeration; CC: Chengdu-Chongqing urban agglomeration; NSTM: Northern Slope of Tianshan Mountains urban agglomeration.

**Comment 2:** Section 2. Subsection 2.2. Please check your weblinks and ensure they are all accessible to the wider scientific community. I could not reach:

- beijingair.sinaapp.com (pollutant data)
- https://data.ess.tsinghua.edu.cn/ (global land cover data)

The latter redirected me to another site, but this hung.

**Response:**

Thanks for your suggestion. The original weblinks have been updated, so they are inaccessible. We have adjusted the two links, and the modified versions are presented as:

https://quotsoft.net/air/ ,

https://data-starcloud.pcl.ac.cn/iearthdata/.

**Comment 3:** Section 2. Subsection 2.3.2. Please could you provide a clearer explanation of the way in which you used GIS techniques to categorize each monitoring station by urban functional zone. I presume you conducted a point-in-polygon test? Did you consider constructing a buffer and assigning the monitoring station to the dominant functional zone within that buffer (to reduce edge effects)?

**Response:**

In our study, overlay analysis is employed to accurately classify monitoring stations into urban functional zones. Given the limited scale of urban functional zones, constructing a buffer zone might encompass multiple functional zones, which could compromise the analysis's precision. After thorough deliberation, we opted not to create a buffer zone and instead proceeded with direct classification. The specific classification method is as follows: We utilize the ArcGIS feature overlay tool to integrate the monitoring station layer with the urban area functional layer. This process generates new feature attributes, which enable us to determine the type of functional area each monitoring station belongs to.

We have added descriptions in the revised manuscript. "In this study, ArcGIS was employed to overlay the latitude and longitude data of the sites with the urban functional area data, allowing for the identification of the functional area category for each site through spatial connections. This process facilitated the addition of new functional area attribute information to the site data." in **Lines 173-176**.

**Comment 4:** Section 3. Subsection 3.1. Please could you provide a new table summarizing the number of monitoring stations per urban functional zone. You could also summarize the number of monitoring stations per urban functional zone per region, and per urban agglomeration. This would be far more effective than the map (Figure

2) which could be deleted.

**Response:**

Thanks for your valuable suggestion. We have deleted Figure 2 and added Table 1 as shown below. In this table, we have summarized the number of stations in each urban functional area based on geographical regions and urban agglomerations. Meanwhile, we have also made modifications to the textual description in **Lines 201-207**: "From the perspective of the six geographical regions, it is evident that the East and South-Central regions contain a relatively high number of stations. To be specific, these two regions are home to 176 and 132 residential zone stations respectively. In contrast, the Northwest region has a significantly smaller number of stations, with only 49 residential zone stations. In terms of the six major urban agglomerations, the Yangtze River Delta urban agglomeration and the Triangle of Central China boast a relatively large number of stations. Conversely, the Northern Slope of Tianshan Mountains urban agglomeration has the fewest stations, with each urban functional zone within it having fewer than 10 stations. Furthermore, among all the urban agglomerations, the Chengdu-Chongqing urban agglomeration has the highest number of transportation zone stations."

Table 1. Statistics on the number of monitoring stations per urban functional zone per region, and per urban agglomeration.

| | | Residential | Commercial | Industrial | Transportation | Public management and service |
|---|---|---|---|---|---|---|
| Six geographical regions | North | 62 | 23 | 35 | 5 | 46 |
| | Northeast | 63 | 15 | 36 | 12 | 32 |
| | East | 176 | 38 | 71 | 13 | 89 |
| | South-central | 132 | 34 | 75 | 22 | 73 |
| | Southwest | 73 | 17 | 25 | 16 | 42 |
| | Northwest | 49 | 15 | 36 | 1 | 38 |
| | Total | 555 | 142 | 278 | 69 | 320 |
| Six urban agglomerations | BTH | 31 | 6 | 12 | 4 | 21 |
| | YRD | 76 | 8 | 27 | 3 | 42 |
| | TC | 66 | 11 | 21 | 2 | 33 |

| | | | | | |
|------|-----|----|----|----|-----|
| GBA | 23 | 3 | 7 | 2 | 19 |
| CC | 37 | 2 | 10 | 7 | 18 |
| NSTM | 9 | 0 | 9 | 1 | 7 |
| Total | 242 | 30 | 86 | 19 | 140 |

**Comment 5:** Section 3. Subsection 3.2.1. Could you come up with a more effective way of showing reductions in pollutant concentrations over time? Your left hand plots show percentage reduction over time and presents all pollutants on similar sized plots but with very different scales on the y-axes (ranging from -8% for $O_3$ to -60% for $SO_2$). Could you either plot these all on the same vertical percentage scale, or, better still, plot on an absolute scale (micrograms per cubic metre) with trend lines. These could usefully a) quantify absolute reductions in pollutant concentrations over time and b) determine whether reductions in some urban functional zones were greater than in others (through an analysis of gradients) which could link to policy measures. Some of this new quantitative information could be included in a revised abstract. Your right hand plots are more difficult to interpret. Are they necessary?

**Response:**

Thanks for your suggestion. In response, we have revised the vertical coordinate of the left graph to represent absolute concentration in micrograms per cubic meter ($\mu g/m^3$). The right graph has been relocated to the supplement to enhance the clarity and coherence of the presentation. Meanwhile, we have conducted trend line fitting for the time series. To prevent the graph from appearing overly intricate due to the direct addition of fitting lines, we have opted to highlight the significance of the relevant data in the main text rather than cluttering the graph, thereby ensuring its clarity and conciseness. For example, "Compared with 2017, $PM_{2.5}$ concentration had decreased by 34.3%, 35.5%, and 33.8% in these areas by 2022, and $SO_2$ concentration had decreased by 55.6%, 56.4%, and 53.4%, respectively ($P < 0.01$)." in **Lines 225-227**.

[Figure]

Figure 2. Annual variation of six pollutant concentrations in various functional zones of Chinese cities. (a) $PM_{2.5}$, (b) $PM_{10}$, (c) $SO_2$, (d) $NO_2$, (e) $O_3$, (f) CO.

**Comment 6:** Section 3. Subsection 3.2.2. (Seasonal). Figure 4 highlights seasonal variations in $PM_{2.5}$ and $O_3$ across the 5 urban functional zones over a 6 year period. I have two comments to make here. Firstly, you only show 2 pollutants, when others, such as $SO_2$ presumably exhibit significant seasonal variation. Why not include these too? Secondly, despite having a separate plot for each urban functional zone, it is difficult to establish whether there are meaningful differences between these zones. Are they all behaving the same, or are some behaving differently. Could you test this,

statistically?

**Response:**

Thank you for your suggestion. In fact, we have comprehensively analyzed the seasonal variations of all six pollutants. Only two representative pollutants are presented in the main text. The seasonal variations of other pollutants can be found in the supplement Figure S2.

We have already conducted a statistical significance test, as shown in Figure S3. The results indicate that, while there exist certain variations in seasonal fluctuations across different urban functional zones, these differences are not statistically significant. Therefore, we have made modifications to the original description: "While there exist certain variations in seasonal fluctuations among different urban functional zones, these differences are not significant (Figure S3)." in **Lines 274-275**.

[Figure]

Figure S3. Statistical significance test of seasonal fluctuations of $PM_{2.5}$ (a), $PM_{10}$ (b), $SO_2$ (c), $NO_2$ (d), $O_3$ (e), and CO (f) in various functional zones of Chinese cities.

**Comment 7:** Section 3. Subsection 3.2.2 (Daily). Figure 5 shows daily variations in pollutant concentrations per urban functional zone and you conclude that some pollutant concentrations vary more in some urban functional zones than others. Could you be more quantitative in your assessment here, please? Could you statistically test for differences? For example, $PM_{2.5}$ looks to exhibit much less diurnal variation in the public management functional zone than other functional zones, but is this statistically significant?

**Response:**

Thank you for your insightful suggestion. We have carried out rigorous statistical tests to analyze the differences in daily variations among various urban functional zones. The results indicate that, with the exception of $NO_2$ and $O_3$, there are significant variations in the mean values of the other pollutants across different functional zones. Therefore, we have incorporated a saliency description into the revised manuscript. "Specifically, the daily variation of $PM_{2.5}$ concentrations in public management and service zone stands at 7.8 µg/m³, significantly lower than that observed in other functional zones ($P < 0.05$). In commercial zone, the daily variation of $SO_2$ and CO notably exceeds that of other functional zones, with levels reaching 6.74 µg/m³ and 0.29 mg/m³, respectively ($P < 0.05$)." in **Lines 298-301**.

[Figure]

Figure 4. Daily variation range of $PM_{2.5}$ (a), $PM_{10}$ (b), $SO_2$ (c), $NO_2$ (d), $O_3$ (e), and CO (f) concentrations in various functional zones of Chinese cities.

**Comment 8:** Section 3. Subsection 3.3.1. Again, I wonder if there are more effective ways of communicating differences over geographical regions and whether these are statistically significant. Could, for example, the bar charts be organised by urban functional zone rather than by geographical region, so we have 6 adjacent bars showing pollutant concentrations for the residential, commercial, industrial, transport and management zones side-by-side? You could label these N, NE, E, SE, SW and NW, respectively. Could these also be tested to determine a) whether concentrations in one zone were significantly different to those in another zone and b) whether concentrations for a single zone, e.g., industry, were significantly different in different geographic regions? You do something similar in Figure 7 which works quite well.

**Response:**

Thank you for your valuable suggestion. We have revised the bar chart by converting its horizontal coordinate into an urban functional area dimension and have also carried out relevant statistical tests. The results indicate that, for a single zone, there

are significant variations in the mean values of the pollutants across different geographic regions. "Furthermore, in terms of industrial zone, the concentrations of various pollutants in Southwest China are significantly lower than those in other regions ($P < 0.05$)."in **Lines 365-366**.

[Figure]

Figure 6. Concentrations of $PM_{2.5}$ (a), $PM_{10}$ (b), $SO_2$ (c), $NO_2$ (d), $O_3$ (e), and CO (f) in each functional zone of the six geographical regions. $*P < 0.05$.

**Comment 9:** Section 3. Subsection 3.3.2. Figure 7 is more effective than Figure 6 and focusses on agglomeration zones not regions. Again, it would help the discussion (management of pollution sources) if differences in pollutant concentrations in different functional zones in different urban agglomerations could be quantified in some way. For example, are $SO_2$ concentrations from industrial sources in BTH and TC significantly different to those in YRD, GBA, CC and NSTM?

**Response:**

Thank you for your suggestion. We have conducted a statistical test for differences in pollutant concentrations among different urban agglomerations. The results indicate

that, there are significant variations in different functional zones in different urban agglomerations. "Furthermore, in terms of industrial zone, the $SO_2$ concentration of BTH obviously surpasses that of other urban agglomerations ($P < 0.01$), reaching 15.6 µg/m³. The industrial structure of BTH is dominated by heavy industry, with industrial land use exhibiting a high degree of concentration, such as steel, chemical industry, building materials, etc. These sectors serve as the primary contributors to $SO_2$ emissions." in **Lines 395-399**. "From the perspective of transportation zone, the $NO_2$ concentration in TC is significantly lower than that in other urban agglomerations ($P < 0.01$), standing at merely 21.2 µg/m³. This is attributed to the open terrain of TC's transportation zone, which facilitates the dispersion of pollutants." in **Lines 402-404**.

[Figure]

Figure 7. Differences in concentrations of $PM_{2.5}$ (a), $PM_{10}$ (b), $SO_2$ (c), $NO_2$ (d), $O_3$ (e), and CO (f) in each functional zone of the six urban agglomerations. BTH: Beijing-Tianjin-Hebei urban agglomeration; YRD: Yangtze River Delta urban agglomeration; TC: Triangle of Central China; GBA: Greater Bay Area urban agglomeration; CC: Chengdu-Chongqing urban agglomeration; NSTM: Northern Slope of Tianshan Mountains urban agglomeration. **$P < 0.01$.

**Comment 10:** Section 3.  Subsection 3.4.1. It is good that you link improvements in

air quality to management actions and here you talk more specifically about 3 key regions which seem different (larger) to those included on Figure 1. I wonder, do we need additional figures at this stage, or could content from line 365 onwards form part of the discussion.

**Response:**

Thanks for your insightful suggestion. In response, we have made a map of the three key areas. For detailed information, please refer to Figure S6 in the supplement.

[Figure]

Figure S6. Three key areas for air pollution control in China: the Beijing-Tianjin-Hebei region and its surrounding areas (BTH-K), the Fen-Wei Plain (FWP), and the Yangtze River Delta region (YRD-K).

**Comment 11:** Section 3. Subsection 3.4.2. I wondered to what extent you were testing altitudinal effects and to what extent you were testing urbanisation effects – with highly populated urban areas and associated infrastructure generally located at lower altitudes. Please check your interpretation of this section.

**Response:**

We performed a Spearman correlation analysis to examine the relationship between population density and altitude. The results revealed a statistically significant

and strong negative correlation between these two variables ($P < 0.001$).

**Comment 12:** Section 3. Subsection 3.4.3. I wondered whether this material would be better placed earlier in the manuscript, maybe after the material on seasonal variations, since this is essentially a story of pollution by time? I also wanted to challenge your meanings of positive and negative weekend effects. For me, less pollution is a positive thing, not a negative thing. Here, I think you interpret 'positive weekend effect' as concentrations being higher than in the week, and vice-versa for 'negative weekend effect'. Again, it would be useful to test whether weekday and weekend differences in pollutant concentrations between different urban functional zones were statistically significant.

**Response:**

Thank you for your insightful suggestion. Taking your suggestion into account, we have adjusted this part after the seasonal variations. Your viewpoint is both valuable and thought-provoking. The reduction of pollution should indeed be a positive phenomenon rather than a negative outcome. Consequently, we have carefully re-examined and comprehensively revised the definition of the 'positive and negative weekend effect'.

"The weekend concentration less than the weekday concentration is defined as the 'positive weekend effect', and the weekend concentration greater than the weekday concentration is defined as the 'negative weekend effect'." in **Lines 313-314**.

After statistical tests, the results show that there are significant differences in pollutant concentrations among various urban functional zones on weekdays and weekends (\*$P < 0.05$).

[Figure]

Figure 5. Weekend effect((weekend-weekday)/weekday) of six pollutant concentrations in various functional zones nationwide. *$P < 0.05$.

**Comment 13:** Section 4.   You do not appear to include a discussion section in your paper.   I would restructure your manuscript such that you have a substantive discussion section that reflects on the main findings of your analysis, comparing and contrasting your results to those reported by others.   I would particularly focus upon the novel element of your study, the use of urban functional zones, and the differential impacts of air quality management on different pollutants in different zones in different regions.   I would also comment on the limitations of your study – for example, you do not really consider scale effects or the issue of variable sample size, with some of your results based on many monitoring sites per urban functional zone and others very few. There is also the issue as to whether you should classify a point based on its immediate interaction with the land use, or some broader geographical zone (e.g., 1km buffer, 5km buffer).

**Response:**

Given that some of the results in this article cover a wide range and contain many detailed analyses, after careful consideration, we believe that if a separate discussion section is added, it may make the overall structure of the article slightly loose. To facilitate readers' understanding, when presenting the results in Section 3, we simultaneously conducted necessary discussions on the relevant contents and adopted

an arrangement method that combines the results with the discussions.

In the revised manuscript, we have incorporated **Section 3.4.4** Policy implications. This addition delves into the differential impacts of urban functional zones on air quality. Based on these insights, we have formulated targeted recommendations aimed at enhancing air quality management.

**3.4.4 Policy implications**

(1) Strengthen the control of air pollution in transportation zone.

The reduction rate of pollutant concentration in transportation zone was considerably lower than that in other regions. This indicates that more stringent air pollution control measures are imperative in transportation-related areas. Regarding vehicle exhaust emission control, it is essential to rigorously enforce vehicle exhaust emission standards. A comprehensive supervision system must be established with reinforced oversight throughout the vehicle exhaust testing process to guarantee that every on-road vehicle consistently complies with emission standards. For vehicles with excessive exhaust emissions, a mandatory maintenance or elimination mechanism should be implemented. This approach aims to reduce exhaust pollutant emissions from motor vehicles at the source and enhance air quality. In addition, to foster the sustainable development of urban transportation, it is necessary to encourage and support the development and extensive application of new energy vehicles (NEVs).

(2) Strengthen the supervision and management of emissions in commercial zone.

The phenomenon of the "weekend effect" of sulfur dioxide in commercial zone serves as a spotlight on the significant influence that emissions from the catering industry exert on air quality. In light of this, it is imperative for the relevant authorities to intensify environmental oversight of catering enterprises. This includes the rigorous enforcement of emission standards, mandating the installation of high-efficiency grease fume purification and waste gas treatment equipment in these enterprises, as well as ensuring the regular maintenance and monitoring of such equipment. Furthermore, in response to the emission peak triggered by the surge in weekend activities within commercial areas, the formulation of a staggered operation policy presents itself as a viable solution. By strategically scheduling the timing of commercial activities, it is possible to prevent

the concentrated release of pollutants. Meanwhile, commercial enterprises should be encouraged to embrace more environmentally friendly business operation models.

(3) Formulate differentiated urban air pollution control strategies.

Significant temporal and spatial variations exist in air pollutant concentrations across different urban functional zones. Therefore, future air pollution control initiatives should integrate air pollution control strategies, address the pollution issues of each functional zone in a coordinated manner, and strengthen cross-regional cooperation and joint prevention efforts. Given the distinct characteristics of individual functional zones, it is essential to develop and execute emission reduction strategies with precision, informed by spatio-temporal differentiation to continuously enhance the process of air quality improvement.

In addition, we have added a discussion of the limitations in the last paragraph of the conclusion in **Lines 594-599**. "However, this study also possesses certain limitations that warrant acknowledgment. The spatial distribution of monitoring stations differs across regions, which may result in a potential loss of variability when data are grouped regionally. Furthermore, this study has only analyzed the influence of several individual factors without conducting a comprehensive evaluation of their relative contributions. In future research, as simulation technologies continue to advance, we will strive to conduct a more thorough analysis of these integrated effects. Additionally, we plan to actively undertake in-depth case studies on representative cities to provide enhanced guidance for urban planning and management."

Given that the vast majority of monitoring stations in China are situated in urban regions, this study adopts a classification method based on the direct interaction between urban functional zones and the stations. However, should the buffer zone method be employed for classification, it might result in the overwriting of multiple functional zones. For instance, in some areas where land use types are complex and unevenly distributed, the land use situation within the buffer zone may differ significantly from the actual situation at the monitoring points.

**Comment 14:** Section 5.    I would encourage you to revisit your conclusions once you

have revised your analysis taking a more critical, quantitative approach to analysing spatial and temporal trends across your different urban functional zones. I do like the method you propose here, and think there is much to be learned from evaluating data grouped across areas with common geographical characteristics to establish which management actions have been most effective at reducing which pollutants across which sectors.

**Response:**

Thanks for your valuable suggestion. We have thoroughly revised the conclusion section of the manuscript. To enhance the depth and accuracy of our analysis, we have incorporated advanced quantitative methods aimed at comprehensively analyzing the spatio-temporal trends inherent in different urban functional zones.

**4 Conclusion**

[revised manuscript text omitted]

In low-altitude regions (below 1000 m), the reduction of these pollutants in transportation zone was minimal. Conversely, in high-altitude regions (above 1500 m), a significant decrease in these pollutants was observed in both residential and industrial zones. The effect of urban scale on pollutant improvement ranges is as follows: in cities with high population density, the differences in the extent of pollutant improvement among functional zones are relatively small. Conversely, in cities with low population density, these differences are more pronounced. Additionally, urban scale has the most significant impact on public management and service zone and the least impact on residential zone.

To sum up, the intricate link between urbanization and air quality, highlighting the need for continuous monitoring and the development of zone-specific air quality strategies. The findings advocate for adaptive urban planning that takes into account the unique challenges posed by urban functional zones and the necessity for innovative pollution mitigation approaches. In essence, the research contributes to a deeper understanding of the complex dynamics of air quality in urban China. It offers valuable guidance for policymakers and urban planners in crafting effective and targeted air quality management strategies, which are essential for achieving sustainable urban environments. The insights gained from this study are not only pertinent to China but also provide a framework for understanding urban air quality challenges and developing appropriate responses in other urban areas globally.

However, this study also possesses certain limitations that warrant acknowledgment. The spatial distribution of monitoring stations differs across regions, which may result in a potential loss of variability when data are grouped regionally. Furthermore, this study has only analyzed the influence of several individual factors without conducting a comprehensive evaluation of their relative contributions. In future research, as simulation technologies continue to advance, we will strive to conduct a more thorough analysis of these integrated effects. Additionally, we plan to actively undertake in-depth case studies on representative cities to provide enhanced guidance for urban planning and management.

---

## Author Comment (AC2)

Dear Editor and Reviewers,

Thank you for your comments concerning our manuscript entitled "Uncovering the Impact of Urban Functional Zones on Air Quality in China". Those comments are all valuable and very helpful for revising and improving our paper, as well as the important guiding significance to our researches. We have studied comments carefully and have made correction which we hope meet with approval. Revisions are indicated within the text using track changes. **All of the responses have been addressed in blue and revisions have been addressed in red as following file of response letter**, and page numbers and line numbers in the response are based on the clean version. In the following, we include a point-by-point response to the comments.

Thank you and best regards.

Yours sincerely,
Wenchao HAN & Yang WANG

**Responses to RC1**

**Comment 1:** The annotation of references is not standardized, for example, "Zhang et al., 2022c" on line 26 should be "Zhang et al., 2022a".

**Response:**

Thanks! We have revised the annotations of the references.

**Comment 2:** This translation of "the Three-Year Action Plan for Winning the Blue Sky Defense Battle" on line 117 and "Suzhou and Qingdao have the most residential zone sites," on line 186, etc. is inappropriate, which is too Chinglish.

**Response:**

Thank you for your suggestion. To ensure a more scientifically accurate translation of the "Three-Year Action Plan for Winning the Blue Sky Defense Battle," we conducted an extensive review of relevant literature. In doing so, we specifically examined three representative papers published in reputable journals: *Environmental Science & Technology*, *Nature Communications*, and *Environmental Science and Ecotechnology*. Therefore, this translation is consistent with the terminology presented in these publications (Liu et al., 2023; Shi et al., 2022; Zheng et al., 2022).

**Reference**

Liu, Y., Geng, G., Cheng, J., Liu, Y., Xiao, Q., Liu, L., Shi, Q., Tong, D., He, K., and Zhang, Q.: Drivers of Increasing Ozone during the Two Phases of Clean Air Actions in China 2013–2020, Environ. Sci. Technol., 57, 8954–8964, https://doi.org/10.1021/acs.est.3c00054, 2023.
Shi, Q., Zheng, B., Zheng, Y., Tong, D., Liu, Y., Ma, H., Hong, C., Geng, G., Guan, D., He, K., and Zhang, Q.: Co-benefits of $CO_2$ emission reduction from China's clean air actions between 2013-2020, Nat Commun, 13, 5061, https://doi.org/10.1038/s41467-022-32656-8, 2022.
Zheng, Y., Xue, T., Zhao, H., and Lei, Y.: Increasing life expectancy in China by achieving its 2025 air quality target, Environmental Science and Ecotechnology, 12, 100203, https://doi.org/10.1016/j.ese.2022.100203, 2022.

**Comment 3:** What is the meaning of this paragraph on line 117-121 in "2.1 Study area"? Why is it placed here?

**Response:**

We mentioned the concept of "three key areas" in **Section 3.4.1**, therefore we provided a relevant introduction within the study area. To enhance clarity and coherence,

these lines have been replaced to **Section 3.4.1**.

**Comment 4:** The production of Figure 1 is not standardized. Because the latitude and longitude grid has already been marked in Figure 1, there is no need for a compass.

**Response:**

Thanks for pointing this out. We have revised Figure 1 in accordance with your feedback and the suggestions provided by another reviewer, as illustrated below.

[Figure]

Figure 1. The location of study area and the distribution map of six geographical regions and six major urban agglomerations. BTH: Beijing-Tianjin-Hebei urban agglomeration; YRD: Yangtze River Delta urban agglomeration; TC: Triangle of Central China; GBA: Greater Bay Area urban agglomeration; CC: Chengdu-Chongqing urban agglomeration; NSTM: Northern Slope of Tianshan Mountains urban agglomeration.

**Comment 5:** "2.3.1 Data preprocessing" should be placed in "2.2 Data sources".

**Response:**

We appreciate your suggestion. For this study, data preprocessing is a crucial step that we believe should be regarded as a distinct component. We have carefully revised and expanded Section 2.3. For more details, please refer to Comment 6.

**Comment 6:** I haven't found any specific data analysis or research methods in "2.3

Data analysis methods".

**Response:**

Thank you for your advice. We would like to clarify that, rather than employing specific data analysis methods, this study places a greater emphasis on data statistics. Therefore, to ensure clarity and avoid any potential ambiguity, we have made the following revisions: "2.3 Data analysis methods" has been removed, the former 2.3.1 has been renumbered to 2.3, and the former 2.3.2 has been renumbered to 2.4. And we have made several specific adjustments to "2.4 Data analysis process". Section 2.4 has been restructured into two distinct subsections: 2.4.1 Overlay Analysis and 2.4.2 Data Statistics. In particular, in "2.4.2 Data Statistics", we have added the formula for calculating the variation trend of pollutant concentrations, which enhances the clarity and completeness of our data interpretation. The details of these revisions can be found on manuscript **Lines 156-193**.

**2.4 Data analysis process**

2.4.1 Overlay Analysis

Overlay analysis is a spatial analysis technique that involves combining multiple geographic layers to reveal spatial relationships and attribute associations between different elements. In general, overlay analysis is mainly used to integrate various types of spatial data, identifying intersecting, union, or difference areas to provide a foundation for subsequent analyses. Common operations include Intersect, Union, Erase, and Spatial join. In this study, ArcGIS was employed to overlay the latitude and longitude data of the sites with the urban functional area data, allowing for the identification of the functional area category for each site through spatial connections. This process facilitated the addition of new functional area attribute information to the site data. Using a similar methodology, the longitude and latitude data were overlaid with DEM grid data to obtain elevation information for each station. This integration of spatial and attribute data not only provided a more nuanced understanding of the spatial distribution of the sites but also laid the groundwork for further investigations into the relationships between site characteristics and urban functional zones.

2.4.2 Data statistics

In this study, MATLAB was employed as the primary computational tool to develop a customized code for batch processing the pre-processed pollutant concentration data obtained from various monitoring sites. This approach facilitated the efficient calculation of daily, monthly, and annual mean values for six key pollutants ($PM_{2.5}$, $PM_{10}$, $SO_2$, $NO_2$, $O_3$, and CO) at each site, thereby providing a comprehensive temporal overview of pollutant concentrations. Following the temporal analysis, the study proceeded to spatially categorize the data based on functional area classifications. The specific operation method is to establish the site index for each functional area. The corresponding pollutant concentration data for each functional area was extracted based on this site index, leading to the classification of pollutant concentration data across different functional zones. Finally, the average concentrations and variation trend of the six pollutants within each functional area were computed across various spatial scales, including six geographical regions, six urban agglomerations, and different altitudes. The variation trend of pollutant concentrations was characterized by the relative rate of decline, with the specific formula presented as follows.

$$Trend\,(\%) = \frac{(x_{2022} - x_{2017}) \div 5}{\sum x_i \div 6} \times 100\%$$

where $i$ represents the year ($i$ = 2017, 2018, ……, 2022), $x$ represents the average concentrations of six pollutants.

**Comment 7:** "3.4 Analysis of influencing factors" only includes three single factor analyses, which is too simple.

**Response:**

The reviewer raises a great point. These three factors are considered of great importance to influence the air pollution of different function zones. The reasons why we focus on governance measures, altitude, and weekdays are as follows: The selection of governance measures, by comparing the three key areas with the other areas, aims to highlight the differential impacts resulting from different management intensities in various regions. There are significant differences in meteorological conditions across different altitude areas, and the scale and development characteristics of cities are closely related to altitude, which has an undeniable impact on the research results.

Moreover, different functional zones have different functions, and there are significant differences in human activities between weekdays and non-weekdays, which directly affect the emission situations of different functional zones.

Following your suggestion, we have added the impact of urban scale, as detailed in Comment 9. In addition, the distribution of functional zones within a city is intricate (as shown in Figure R1), and there may be interactions between different functional zones. To accurately quantify the relative contributions of various factors, extremely refined model simulations are required, but this work still faces many challenges at present. This is expected to be a remarkably valuable and important subject of study in the future. Your suggestion is indeed valid, prompting us to incorporate a limitation into our discussion (**Lines 595-598**): "Furthermore, this study has only analyzed the influence of several individual factors without conducting a comprehensive evaluation of their relative contributions. In future research, as simulation technologies continue to advance, we will strive to conduct a more thorough analysis of these integrated effects."

[Figure]

Figure R1. Distribution of urban functional zones.

**Comment 8:** Overall, this manuscript contains too much data description and lacks data analysis, especially discussion.

**Response:**

We agree with your valuable comment. In the manuscript, several sections did lack sufficient elaboration. As suggested, we have incorporated additional explanations and quantification description in the discussion to address these deficiencies.

[revised manuscript text omitted]

**Comment 9:** When analyzing the impact of different urban functional zones on air quality, the authors overlooked the scale effect of different functional zones, especially the scale effect of various level cities.

**Response:**

We highly appreciate this great advice. Considering your comment, we have incorporated a new **Section 3.4.3**, which explores the impact of urban scale. Urban scale can be categorized based on several indicators, including economic output, geographical area, and population size. Nevertheless, there is a high degree of correlation among these indicators. For example, cities with robust economic development tend to have larger geographical areas and higher population densities. When it comes to reflecting the impact of air pollution on health, population density has an advantage over economic and area indicators. It can more directly reveal the extent of the harm caused by air pollution to people's health. After careful consideration of all the above factors, we have ultimately decided to use population density as the criterion to differentiate the scale of a city. The population data utilized in this study is sourced from the LandScan Global dataset, which was developed by ORNL. We have incorporated a more comprehensive presentation of the data **in Lines 151-155** of the manuscript. In addition, we have added a description of the urban scale into both the conclusion and the abstract.

**3.4.3 The impact of urban scale**

The urban scale was categorized according to population density, and comparisons were made between high-density and low-density cities regarding the variations in pollutant concentrations across different functional zones (Figure 10). Specifically, citys with a population density exceeding 510 persons/km² (the 70th percentile) were defined as high-density, whereas those with a density below 151 persons/km² (the 30th percentile) were considered low-density.

The results indicate that cities with high population density exhibit significantly higher concentrations of pollutants compared to those with low population density. In cities with low population density, the improvement in particulate matter ($PM_{2.5}$ and $PM_{10}$) levels in public management and service zone is minimal. In addition, the reduction of $SO_2$, $NO_2$, and CO is more pronounced in residential and commercial zones, while the improvement in transportation zone is the least significant. In cities with high population density, the improvement in $PM_{2.5}$ and $SO_2$ concentrations in transportation zone is notably smaller compared to other functional zones, at 8.0% and 11.4%, respectively. This can be attributed to the relatively fixed sources of pollutant emissions in transportation zone, particularly in high population density urban areas where traffic flow remains challenging to significantly reduce even with optimized traffic management, thereby posing greater difficulty in achieving improvements (Lopez-Aparicio et al., 2025). Additionally, the high proportion of impervious pavement in transportation zone contributes to the "heat island effect," which hinders the dispersion of pollutants (Yuan et al., 2018). Therefore, future efforts should prioritize pollution control in transportation zone.

From the perspective of different functional zones, the differences of variation in pollutant levels among urban regions with high population density is less pronounced compared to those with low-density (with the exception of $SO_2$ and $O_3$). When examining the same functional area across cities of varying population densities, significant differences are observed in public management and service zone, where $PM_{2.5}$ and $PM_{10}$ levels differ by 4.5% and 3.8%, respectively. In contrast, residential zone exhibits minimal variation, with $PM_{2.5}$ and $PM_{10}$ levels differing by only 0.2% and

1.0% (Table S1).

[Figure]

Figure 10. Annual variation trend of $PM_{2.5}$ (a), $PM_{10}$ (b), $SO_2$ (c), $NO_2$ (d), $O_3$ (e), and CO (f) concentrations in various functional zones in different urban scale.

Table S1. Variation differences in pollutant concentrations among different urban scale (low density-high density).

|  | $PM_{2.5}$ | $PM_{10}$ | $SO_2$ | $NO_2$ | $O_3$ | CO |
|---|---|---|---|---|---|---|
| Residential | 0.23% | 1.01% | 4.17% | 0.19% | -1.28% | -0.43% |
| Commercial | 2.33% | 1.66% | 4.65% | 1.75% | -1.53% | 1.22% |
| Industrial | 1.76% | 1.25% | 6.26% | 2.10% | -0.92% | 0.93% |
| Transportation | -0.33% | 1.63% | 2.73% | 4.59% | -0.33% | 1.10% |
| Public management and service | 4.47% | 3.77% | 5.28% | 3.88% | -2.04% | 0.39% |

**Response:**

Thanks. We have removed this sentence. Furthermore, in light of the deficiency in specific quantitative data within the abstract, we have incorporated several quantitative metrics to enhance its informativeness.

This study presents a comprehensive spatiotemporal analysis of air quality across various urban functional zones in China from 2017 to 2022, uncovering distinct impacts on air quality due to the unique characteristics of each zone. A general decrease in various pollutant concentrations is observed, a result of stringent pollution control policies. Specifically, the concentration of $PM_{2.5}$ decreased from 46.1 μg/m³ to 30.6 μg/m³. Residential, commercial, and industrial zones show significant declines, whereas the transportation zone experiences the least decrease. However, ozone levels rebound significantly in densely populated residential and commercial zones, and exhibit distinct weekend effects. The research highlights U-shaped seasonal patterns for five key pollutants and inverse seasonal patterns for ozone, which gradually decrease. Furthermore, the daily and seasonal variations of pollutant concentrations in industrial zone are the largest, while those in the public management and service zone are the smallest. For example, the seasonal fluctuation of $PM_{2.5}$ and $PM_{10}$ in industrial zone was 50.5 μg/m³ and 66.1 μg/m³, respectively. Urban scale has the most significant impact on public management and service zone. Notably, spatial heterogeneity is evident, with regional pollutant distributions linked to local emissions, control measures, urban morphology, and climate variability. This study emphasizes the critical link between urbanization and air quality, advocating for continuous monitoring and the development of zone-specific air quality strategies to ensure sustainable urban environments.

---

## Author Comment (AC4)

Dear Editor and Reviewers,

Thank you for your comments concerning our manuscript entitled "Uncovering the Impact of Urban Functional Zones on Air Quality in China". Those comments are all valuable and very helpful for revising and improving our paper, as well as the important guiding significance to our researches. We have studied comments carefully and have made correction which we hope meet with approval. Revisions are indicated within the text using track changes. **All of the responses have been addressed in blue and revisions have been addressed in red as following file of response letter**, and page numbers and line numbers in the response are based on the clean version. In the following, we include a point-by-point response to the comments.

Thank you and best regards.

Yours sincerely,
Wenchao HAN & Yang WANG

**Responses to CC**

**Comment:** In this study, the concentration of air pollutants in different urban functional areas was counted by means of cartographic classification criteria. This novel idea is of great significance to the study of urban air pollution. In the future, the author can consider carrying out specific case studies for typical cities and expanding management experience.

**Response:**

Thank you very much for your valuable suggestion. We fully agree that specific case studies for typical cities and expanding management experience are important directions for future research. In our subsequent work, we will actively consider conducting in-depth case analyses on representative cities to gain more practical insights.

We have added a discussion of the limitations in the last paragraph of the conclusion in **Lines 598-599**. "Additionally, we plan to actively undertake in-depth case studies on representative cities to provide enhanced guidance for urban planning and management."